# A Note on "Assessing Generalization of SGD via Disagreement"

**Andreas Kirsch**      *andreas.kirsch@cs.ox.ac.uk*
**Yarin Gal**      *yarin.gal@cs.ox.ac.uk*
*OATML, Department of Computer Science*
*University of Oxford*

**Reviewed on OpenReview:** *https://openreview.net/forum?id=oRP8urZ8Fx*

## Abstract

Several recent works find empirically that the average test error of deep neural networks can be estimated via the prediction disagreement of models, which does not require labels. In particular, Jiang et al. (2022) show for the disagreement between two separately trained networks that this 'Generalization Disagreement Equality' follows from the well-calibrated nature of deep ensembles under the notion of a proposed 'class-aggregated calibration.' In this reproduction, we show that the suggested theory might be impractical because a deep ensemble's calibration can deteriorate as prediction disagreement increases, which is precisely when the coupling of test error and disagreement is of interest, while labels are needed to estimate the calibration on new datasets[1]. Further, we simplify the theoretical statements and proofs, showing them to be straightforward within a probabilistic context, unlike the original hypothesis space view employed by Jiang et al. (2022).

## 1 Introduction

Machine learning models can cause harm when their predictions become unreliable, yet we trust them blindly. This is not fiction but has happened in real-world applications of machine learning (Schneider, 2021). Thus, there has been significant research interest in model robustness, uncertainty quantification, and bias mitigation. In particular, finding ways to bound the test error of a trained deep neural network without access to the labels would be of great importance: one could estimate the performance of models in the wild where unlabeled data is ubiquitous, labeling is expensive, and the data often does not match the training distribution. Crucially, it would provide a signal on when to trust the output of a model and when to defer to human experts instead.

Several recent works (Chen et al., 2021; Granese et al., 2021; Garg et al., 2022; Jiang et al., 2022) look at the question of how model predictions in a non-Bayesian setting can be used to estimate model accuracy. In this paper, we focus on one work[2] (Jiang et al., 2022) specifically, and examine the theoretical and empirical results from a Bayesian perspective.

In a Bayesian setting, *epistemic uncertainty* (Der Kiureghian & Ditlevsen, 2009) captures the uncertainty of a model about the reliability of its predictions, that is epistemic uncertainty quantifies the uncertainty of a model about its predictive distribution, while *aleatoric uncertainty* quantifies the ambiguity within the predictive distribution and the label noise, c.f. Kendall & Gal (2017). Epistemic uncertainty thus tells us whether we can trust a model's predictions or not. Assuming a *well-specified* and *well-calibrated* Bayesian model, when its predictive distribution has low epistemic uncertainty for an input, it can be trusted. But likewise, a Bayesian model's calibration ought to deteriorate as epistemic uncertainty increases for a sample: in that case, the predictions become less reliable and so does the model's calibration.

---

[1]Code at https://github.com/BlackHC/2202.01851
[2]An ICLR 2022 Spotlight, which has spawned additional follow-up works, e.g. Baek et al. (2022).

In this context, calibration is an *aleatoric* metric for a model's reliability (Gopal, 2021). Calibration captures how well a model's confidence for a given prediction matches the actual frequency of that prediction in the limit of observations *in distribution*: when a model is 70% confident about assigning label *A*, does *A* indeed occur with 70% probability in these instances?

Jiang et al. (2022), while not Bayesian, make the very interesting empirical and theoretical discovery that deep ensembles satisfy a '*Generalization Disagreement Equality*' when they are well-calibrated according to a proposed '*class-aggregated calibration*' (or a '*class-wise calibration*') and empirically find that the respective calibration error generally bounds the absolute difference between the test error and '*disagreement rate*.' Jiang et al. (2022)'s theory builds upon Nakkiran & Bansal (2020)'s '*Agreement Property*' and provides backing for an empirical connection between the *test error* and *disagreement rate* of two separately trained networks on the same training data. Yet, while Nakkiran & Bansal (2020) limit the applicability of their Agreement Property to in-distribution data, Jiang et al. (2022) carefully extend it: '*our theory is general and makes no restrictions on the hypothesis class, the algorithm, the source of stochasticity, or the test distributions (which may be different from the training distribution)*' with qualified evidence: '*we present preliminary observations showing that GDE is approximately satisfied even for certain distribution shifts within the PACS (Li et al., 2017) dataset.*'

In this paper, we present a new perspective on the theoretical results using a standard probabilistic approach for discriminative (Bayesian) models, whereas Jiang et al. (2022) use a hypothesis space of models that output one-hot predictions. Indeed, their theory does not require one-hot predictions, separately trained models (deep ensembles) or Bayesian models. Moreover, as remarked by the authors, their theoretical results also apply to a single model that outputs softmax probabilities. We will see that our perspective greatly simplifies the results and proofs.

This also means that the employed notion of disagreement rate *does not capture epistemic uncertainty but overall uncertainty*, similar to the predictive entropy, which is a major difference to Bayesian approaches which can evaluate epistemic uncertainty separately (Smith & Gal, 2018). Overall uncertainty is the sum of aleatoric and epistemic uncertainty. For in-distribution data, epistemic uncertainty will generally be low: overall uncertainty will mainly capture aleatoric uncertainty and align well with (aleatoric) calibration measures. However, under distribution shift, epistemic uncertainty can be a confounding factor.

Importantly, we find that the connection between the proposed calibration metrics and the gap between test error and disagreement rate exists because the introduced notion of class-aggregated calibration is so strong that this connection follows almost at once.

Moreover, the suggested approach is circular[3]: calibration must be measured on the data distribution we want to evaluate. Otherwise, we cannot bound the difference between the test error and the disagreement rate and obtain a signal on how trustworthy our model is. This reintroduces the need for labels on the unlabeled dataset, limiting practicality. Alternatively, one would have to assume that these calibration metrics do not change for different datasets or under distribution shifts, which we show not to hold: deep ensembles are less calibrated the more the ensemble members disagree (even on in-distribution data).

Lastly, we draw connections and show that the 'class-aggregated calibration error' and the 'class-wise calibration error'[4] are equivalent to the 'adaptive calibration error' and 'static calibration error' introduced in Nixon et al. (2019) and its implementation.

**Outline.** We introduce the necessary background and notation in §2. In §3 we rephrase the theoretical statements from Jiang et al. (2022) using a parameter distribution (instead of a version space) and auxiliary random variables. This allows us to simplify the theoretical statements and proofs greatly in §4 and to examine the connection to Nixon et al. (2019). Finally, in §5, we provide empirical evidence that deep ensembles are less calibrated exactly when their ensemble members disagree.

---

[3]This was added as a caveat to the camera-ready version of Jiang et al. (2022) after reviewing a preprint of this paper.
[4]Which is not explicitly introduced in Jiang et al. (2022) but can be analogously constructed.

## 2 Background & Setting

We introduce relevant notation, the initial Bayesian formalism, the connection to deep ensembles, and the probabilistic model. We restate the statements from Jiang et al. (2022) using this formalism in §3.

**Notation.** We use an implicit notation for expectations $\mathbb{E}[f(X)]$ when possible. For additional clarity, we also use $\mathbb{E}_X[f(X)]$ and $\mathbb{E}_{\mathrm{p}(x)} f(x)$, which fix the random variables and distribution, respectively, when needed.

We will use nested probabilistic expressions of the form $\mathbb{E}[\mathrm{p}(\hat{Y} = Y \mid X)]$. Prima facie, this seems unambiguous, but is $\mathrm{p}(\hat{Y} = Y \mid X)$ a transformed random variable of only $X$ or also of $Y$ (and $\hat{Y}$): what are we taking the expectation over? This is not always unambiguous, so we disambiguate between the probability for an event defined by an expression $\mathbb{P}[\ldots] = \mathbb{E}[\mathbb{1}\{\ldots\}]$, where $\mathbb{1}\{\ldots\}$ is the indicator function[5], and a probability given specific outcomes for various random variables $\mathrm{p}(\hat{y} \mid x)$, c.f.:

$$\mathbb{P}[\hat{Y} = Y \mid X] = \mathbb{E}_{\hat{Y}, Y}[\mathbb{1}\{\hat{Y} = Y\} \mid X] = \mathbb{E}_{\mathrm{p}(\hat{y}, y \mid X)} \mathbb{1}\{\hat{y} = y\}, \tag{1}$$

which is a transformed random variable of $X$, while $\mathrm{p}(\hat{Y} = Y \mid X)$ is simply a (transformed) random variable, applying the probability density on the random variables $Y$ and $X$. Put differently, $Y$ is bound within the former but not the latter: $\mathbb{P}[\ldots \mid X]$ is a transformed random variable of $X$, and any random variable that appears within the $\ldots$ is bound within that expression.

**Probabilistic Model.** We assume classification with $K$ classes. For inputs $X$ with ground-truth labels $Y$, we have a Bayesian model with parameters $\Omega$ that makes predictions $\hat{Y}$:

$$\mathrm{p}(y, \hat{y}, \omega \mid x) = \mathrm{p}(y \mid x)\, \mathrm{p}(\hat{y} \mid x, \omega)\, \mathrm{p}(\omega). \tag{2}$$

We focus on model evaluation. (Input) samples $x$ can come either from 'in-distribution data' which follows the training set or from samples under covariate shift (distribution shift). The expected prediction over the model parameters is the *marginal predictive distribution*:

$$\mathrm{p}(\hat{y} \mid x) = \mathbb{E}_{\Omega}[\mathrm{p}(\hat{y} \mid x, \Omega)]. \tag{3}$$

**On $\mathrm{p}(\omega)$.** The main emphasis in Bayesian modelling can be Bayesian inference or Bayesian model averaging (Wilson & Izmailov, 2020). Here we concentrate on the model averaging perspective, and for simplicity take the model averaging to be with respect to *some* distribution $\mathrm{p}(\omega)$. Hence, we will use $\mathrm{p}(\omega)$ as the push-forward of models initialized with different initial seeds through SGD to minimize the negative log likelihood with weight decay and a specific learning rate schedule (MLE or MAP) (Mukhoti et al., 2021):

**Assumption 1.** We assume that $\mathrm{p}(\omega)$ is a distribution of possible models we obtain by training with a specific training regime on the training data with different seeds. A single $\omega$ identifies a single trained model.

**Deep Ensembles.** We cast deep ensembles (Hansen & Salamon, 1990; Lakshminarayanan et al., 2016), which refer to training multiple models and averaging predictions, into the introduced Bayesian perspective above by viewing them as an empirical finite sample estimate of the parameter distribution $\mathrm{p}(\omega)$. Then, $\omega_1, \ldots, \omega_N \sim \mathrm{p}(\omega)$ drawn i.i.d. are the *ensemble members*.

Again, the implicit model parameter distribution $\mathrm{p}(w)$ is given by the models that are obtained through training. Hence, we can view the predictions of a deep ensemble or the ensemble's prediction disagreement for specific $x$ (or over the data) as empirical estimates of the predictions or the model disagreement using the implicit model distribution, respectively.

**Calibration.** A model's calibration for a given $x$ measures how well the model's *top-1 (argmax) confidence*

$$\mathrm{Conf}_{\mathrm{Top1}} := \mathrm{p}(\hat{Y} = \arg\max_k \mathrm{p}(\hat{Y} = k \mid X) \mid X) \tag{4}$$

matches its *top-1 accuracy*

$$\mathrm{Acc}_{\mathrm{Top1}} := \mathrm{p}(Y = \arg\max_k \mathrm{p}(\hat{Y} = k \mid X) \mid X), \tag{5}$$

---

[5]The indicator function is 1 when the predicate '$\ldots$' is true and 0 otherwise.

where we define both as transformed random variables of $X$. The calibration error is usually defined as the absolute difference between the two:

$$\text{CE} := |\text{Acc}_{\text{Top1}} - \text{Conf}_{\text{Top1}}|. \tag{6}$$

In general, we are interested in the *expected calibration error (ECE)* over the data distribution (Guo et al., 2017) where we bin samples by their top-1 confidence. Intuitively, the ECE will be low when we can trust the model's top-1 confidence on the given data distribution.

We usually use top-1 predictions in machine learning. However, if we were to draw $\hat{Y}$ according to $\text{p}(\hat{y} \,|\, x)$ instead, the (expected) accuracy would be:

$$\text{Acc} := \mathbb{P}[Y = \hat{Y} \,|\, X] \tag{7}$$

$$= \sum_k \text{p}(Y = k \,|\, X)\, \text{p}(\hat{Y} = k \,|\, X) \tag{8}$$

$$= \mathbb{E}_Y[\text{p}(\hat{Y} = Y \,|\, X) \,|\, X], \tag{9}$$

as a random variable of $X$. Usually we are interested in the accuracy over the whole dataset:

$$\mathbb{P}[\hat{Y} = Y] = \mathbb{E}[\text{Acc}] = \mathbb{E}_X[\mathbb{P}[\hat{Y} = Y \,|\, X]] = \mathbb{E}_{X,Y}[\text{p}(\hat{Y} = Y \,|\, X)]. \tag{10}$$

For example, for binary classification with two classes A and B, if class A appears with probability 0.7 and a model predicts class A with probability 0.2 (and thus class B appears with probability 0.3, which a model predicts as 0.8), its accuracy is $0.7 \times 0.2 + 0.3 \times 0.8 = 0.38$, while the top-1 accuracy is 0.3. Likewise, the predicted accuracy is $0.2^2 + 0.8^2 = 0.68$ while the top-1 predicted accuracy is 0.8.

## 3 Rephrasing Jiang et al. (2022) in a Probabilistic Context

We present the same theoretical results as Jiang et al. (2022) but use a Bayesian formulation instead of a hypothesis space and define the relevant quantities as (transformed) random variables. As such, our definitions and theorems are equivalent and follow the paper but look different. We show these equivalences in §A in the appendix and prove the theorems and statements themselves in the next section.

First, however, we note a distinctive property of Jiang et al. (2022). It is assumed that each $\text{p}(\hat{y} \,|\, x, \omega)$ is always one-hot for any $\omega$. In practice, this could be achieved by turning a neural network's softmax probabilities into a one-hot prediction for the $\arg\max$ class. We call this the *Top1-Output-Property* (TOP).

**Assumption 2.** The Bayesian model $\text{p}(\hat{y}, \omega \,|\, x)$ satisfies TOP: $\text{p}(\hat{y} \,|\, x, \omega)$ is one-hot for all $x$ and $\omega$.

**Definition 3.1.** The *test error* and *disagreement rate*, as transformed random variables of $\Omega$ (and $\Omega'$), are:

$$\text{TestError} := \mathbb{P}[\hat{Y} \neq Y \,|\, \Omega] \tag{11}$$

$$\big( = 1 - \mathbb{P}[\hat{Y} = Y \,|\, \Omega] \tag{12}$$

$$= 1 - \mathbb{E}_{X,Y}[\text{p}(\hat{Y} = Y \,|\, X, \Omega)], \tag{13}$$

$$= 1 - \mathbb{E}_{\text{p}(x,y)}\, \text{p}(\hat{Y} = y \,|\, x, \Omega))\big), \tag{14}$$

$$\text{Dis} := \mathbb{P}[\hat{Y} \neq \hat{Y}' \,|\, \Omega, \Omega'] \tag{15}$$

$$\big( = 1 - \mathbb{P}[\hat{Y} = \hat{Y}' \,|\, \Omega, \Omega'] \tag{16}$$

$$= 1 - \mathbb{E}_{X,\hat{Y}}[\text{p}(\hat{Y}' = \hat{Y} \,|\, X, \Omega') \,|\, \Omega, \Omega'] \tag{17}$$

$$= 1 - \mathbb{E}_{\text{p}(x,\hat{y}|\Omega)}\, \text{p}(\hat{Y}' = \hat{y} \,|\, x, \Omega'))\big), \tag{18}$$

where for the disagreement rate, we expand our probabilistic model to take a second model $\Omega'$ with prediction $\hat{Y}'$ into account (and which uses the same parameter distribution), so:

$$\text{p}(y, \hat{y}, \omega, \hat{y}', \omega' \,|\, x) := \text{p}(y \,|\, x)\, \text{p}(\hat{y} \,|\, x, \omega)\, \text{p}(\omega)\, \text{p}(\hat{Y} = \hat{y}' \,|\, x, \Omega = \omega')\, \text{p}(\Omega = \omega').$$

Jiang et al. (2022) then introduce the property of interest:

**Definition 3.2.** A Bayesian model $p(\hat{y}, \omega \mid x)$ fulfills the *Generalization Disagreement Equality (GDE)* when:

$$\mathbb{E}_\Omega[\text{TestError}(\Omega)] = \mathbb{E}_{\Omega, \Omega'}[\text{Dis}(\Omega, \Omega')] \quad (\Leftrightarrow \mathbb{E}[\text{TestError}] = \mathbb{E}[\text{Dis}]). \tag{19}$$

When this property holds, we seemingly do not require knowledge of the labels to estimate the (expected) test error: computing the (expected) disagreement rate is sufficient.

Two different types of calibration are then introduced, *class-wise* and *class-aggregated* calibration, and it is shown that they imply the GDE:

**Definition 3.3.** The Bayesian model $p(\hat{y}, \omega \mid x)$ satisfies *class-wise calibration* when for any $q \in [0, 1]$ and any class $k \in [K]$:

$$p(Y = k \mid p(\hat{Y} = k \mid X) = q) = q. \tag{20}$$

Similarly, the Bayesian model $p(\hat{y}, \omega \mid x)$ satisfies *class-aggregated calibration* when for any $q \in [0, 1]$:

$$\sum_k p(Y = k, p(\hat{Y} = k \mid X) = q) = q \sum_k p(p(\hat{Y} = k \mid X) = q). \tag{21}$$

**Theorem 3.4.** *When the Bayesian model $p(\hat{y}, \omega \mid x)$ satisfies class-wise or class-aggregated calibration, it also satisfies GDE.*

Finally, Jiang et al. (2022) introduce the *class-aggregated calibration error* similar to the ECE and then use it to bound the magnitude of any GDE gap:

**Definition 3.5.** The *class-aggregated calibration error (CACE)* is the integral of the absolute difference of the two sides in eq. (21) over possible $q \in [0, 1]$:

$$\text{CACE} := \int_{q \in [0,1]} \left| \sum_k p(Y = k, p(\hat{Y} = k \mid X) = q) - q \sum_k p(p(\hat{Y} = k \mid X) = q) \right| dq. \tag{22}$$

**Theorem 3.6.** *For any Bayesian model $p(\hat{y}, \omega \mid x)$, we have:*

$$|\mathbb{E}[\text{TestError}] - \mathbb{E}[\text{Dis}]| \leq \text{CACE}.$$

In the following section, we simplify the definitions and prove the statements using elementary probability theory, showing that notational complexity is the main source of complexity.

## 4 GDE is Class-Aggregated Calibration in Expectation

We show that proof for Theorem 3.6 is trivial if we use different but equivalent definitions of the class-wise and class-aggregate calibration. First though, we establish a better understanding for these definitions by examining the GDE property $\mathbb{E}[\text{TestError}] = \mathbb{E}[\text{Dis}]$. For this, we expand the definitions of $\mathbb{E}[\text{TestError}]$ and $\mathbb{E}[\text{Dis}]$, and use random variables to our advantage.

We define a quantity which will be of intuitive use later on: the *predicted accuracy*

$$\text{PredAcc} := \mathbb{E}_{\hat{Y}}[p(\hat{Y} \mid X) \mid X] = \sum_k p(\hat{Y} = k \mid X) \, p(\hat{Y} = k \mid X), \tag{23}$$

as a random variable of $X$. It measures the expected accuracy assuming the model's predictions are correct, that is the true labels follow $p(\hat{y} \mid x)$. This also assumes that we draw $\hat{Y}$ accordingly and do not always use the top-1 prediction.

**Revisiting GDE.** On the one hand, we have:

$$\mathbb{E}[\text{TestError}] = \mathbb{E}_\Omega[\mathbb{P}[\hat{Y} \neq Y \mid \Omega]] \tag{24}$$

$$= 1 - \mathbb{P}[\hat{Y} = Y] \tag{25}$$

$$= 1 - \mathbb{E}_{X,\hat{Y}}[\mathrm{p}(Y = \hat{Y} \mid X)] \tag{26}$$

$$= 1 - \mathbb{E}[\mathrm{Acc}] \tag{27}$$

and on the other hand, we have:

$$\mathbb{E}[\mathrm{Dis}] = \mathbb{E}_{\Omega,\Omega'}[\mathbb{P}[\hat{Y} \neq \hat{Y}' \mid \Omega, \Omega']] \tag{28}$$

$$= 1 - \mathbb{E}_{\Omega,\Omega'}[\mathbb{P}[\hat{Y} = \hat{Y}' \mid \Omega, \Omega']] \tag{29}$$

$$= 1 - \mathbb{P}[\hat{Y} = \hat{Y}'] \tag{30}$$

$$= 1 - \mathbb{E}_{X,\hat{Y}}[\mathrm{p}(\hat{Y}' = \hat{Y} \mid X)] \tag{31}$$

$$= 1 - \mathbb{E}_{X,\hat{Y}}[\mathrm{p}(\hat{Y} \mid X)] \tag{32}$$

$$= 1 - \mathbb{E}[\mathrm{PredAcc}]. \tag{33}$$

The step from (30) to (31) is valid because $\hat{Y} \perp\!\!\!\perp \hat{Y}' \mid X$, and the step from (31) to (32) is valid because $\mathrm{p}(\hat{y}' \mid x) = \mathrm{p}(\hat{y} \mid x)$. Thus, we can rewrite Theorem 3.4 as:

> **Lemma 4.1.** *The model* $\mathrm{p}(\hat{y} \mid x)$ *satisfies GDE, when*
>
> $$\mathbb{E}[\mathrm{Acc}] = \mathbb{E}[\mathrm{p}(Y = \hat{Y} \mid X)] = \mathbb{E}[\mathrm{p}(\hat{Y} \mid X)] = \mathbb{E}[\mathrm{PredAcc}], \tag{34}$$
>
> *i.e. in expectation, the accuracy of the model equals the predicted accuracy of the model, or equivalently, the error of the model equals its predicted error.*

Crucially, while Jiang et al. (2022) calls $1 - \mathbb{E}_{X,\hat{Y}}[\mathrm{p}(\hat{Y} \mid X)]$ the (expected) disagreement rate $\mathbb{E}[\mathrm{Dis}]$, it actually is just the predicted error of the (Bayesian) model as a whole.

Equally important, all dependencies on $\Omega$ have vanished. Indeed, we will not use $\Omega$ anymore for the remainder of this section. This reproduces the corresponding remark from Jiang et al. (2022)[6]:

> *Conclusion* 1. The theoretical statements in Jiang et al. (2022) can be made about any discriminative model with predictions $\mathrm{p}(y \mid x)$.

When is $\mathbb{E}_{X,\hat{Y}}[\mathrm{p}(Y = \hat{Y} \mid X)] = \mathbb{E}_{X,\hat{Y}}[\mathrm{p}(\hat{Y} \mid X)]$? Or in other words: when does $\mathrm{p}(Y = \hat{y} \mid x)$ equal $\mathrm{p}(\hat{Y} = \hat{y} \mid x)$ in expectation over $\mathrm{p}(x, y, \hat{y})$?

As a trivial sufficient condition, when the predictive distribution matches our data distribution—*i.e. when the model* $\mathrm{p}(\hat{y} \mid x)$ *is perfectly calibrated on average for all classes—and not only for the top-1 predicted class.* $ECE = 0$ is not sufficient because the standard calibration error only ensures that the data distribution and predictive distribution match for the top-1 predicted class (Nixon et al., 2019). But class-wise calibration entails this equality.

**Class-Wise and Class-Aggregated Calibration.** To see this, we rewrite class-wise and class-aggregated calibration slightly by employing the following tautology:

$$\mathrm{p}(\hat{Y} = k \mid \mathrm{p}(\hat{Y} = k \mid X) = q) = q, \tag{35}$$

which is obviously true due its self-referential nature. We provide a formal proof in §D in the appendix. Then we have the following equivalent definition:

**Lemma 4.2.** *The model* $\mathrm{p}(\hat{y} \mid x)$ *satisfies* class-wise calibration *when for any* $q \in [0, 1]$ *and any class* $k \in [K]$:

$$\mathrm{p}(Y = k, \mathrm{p}(\hat{Y} = k \mid X) = q) = \mathrm{p}(\hat{Y} = k, \mathrm{p}(\hat{Y} = k \mid X) = q). \tag{36}$$

---

[6]The remark did not exist in the first preprint version.

*Similarly, the model* $p(\hat{y} \mid x)$ *satisfies* class-aggregated calibration *when for any* $q \in [0, 1]$:

$$p(p(\hat{Y} = Y \mid X) = q) = p(p(\hat{Y} \mid X) = q), \tag{37}$$

*and* class-wise *calibration implies* class-aggregate *calibration.*

The straightforward proof is found in §D in the appendix.

Jiang et al. (2022) mention 'level sets' as intuition in their proof sketch. Here, we have been able to make this even clearer: class-aggregated calibration means that level-sets for accuracy $p(\hat{Y} = Y \mid X)$ and predicted accuracy $p(\hat{Y} \mid X)$—as random variables of $Y$ and $X$, and $\hat{Y}$ and $X$, respectively—have equal measure, that is probability, for all $q$.

---

**GDE.** Now, class-aggregated calibration immediately and trivially implies GDE. To see this, we use the following property of expectations:

**Lemma 4.3.** *For a random variable* $X$, *a function* $t(x)$, *and the random variable* $T = t(X)$, *it holds that*

$$\mathbb{E}_T[T] = \mathbb{E}[T] = \mathbb{E}_X[t(X)]. \tag{38}$$

This basic property states that we can either compute an expectation over $T$ by integrating over $p(T = t)$ or by integrating $t(x)$ over $p(X = x)$. This is just a change of variable (push-forward of a measure).

We can use this property together with the class-aggregated calibration to see:

$$
\begin{array}{cc}
\mathbb{E}[\mathrm{Acc}] & \mathbb{E}[\mathrm{PredAcc}] \\
\| & \| \\
\mathbb{E}_{X,Y}[p(\hat{Y} = Y \mid X)] & \mathbb{E}_{X,\hat{Y}}[p(\hat{Y} \mid X)] \\
\| & \| \\
\mathbb{E}[p(\hat{Y} = Y \mid X)] & \mathbb{E}[p(\hat{Y} \mid X)] \\
\| & \| \\
\mathbb{E}_{q \sim p(\hat{Y}=Y \mid X)}[q] & = \quad \mathbb{E}_{q \sim p(\hat{Y} \mid X)}[q] \quad,
\end{array}
\tag{39}
$$

which is exactly Lemma 4.1, where we start with the equality following from class-aggregated calibration and then apply Lemma 4.3 along each side. Thus, GDE is but an expectation over class-aggregated calibration; we have:

**Theorem 4.4.** *When a model* $p(\hat{y} \mid x)$ *satisfies class-wise or class-aggregated calibration, it satisfies GDE.*

---

*Proof.* We can formalize the proof to be even more explicit and introduce two auxiliary random variables:

$$S := p(\hat{Y} = Y \mid X), \tag{40}$$

as a transformed random variable of $Y$ and $X$, and

$$T := p(\hat{Y} \mid X), \tag{41}$$

as a transformed random variable of $\hat{Y}$ and $X$. Class-wise calibration implies class-aggregated calibration. Class-aggregated calibration then is $p(S = q) = p(T = q)$ (*). Writing out eq. (39), we have

$$\mathbb{E}[p(\hat{Y} = Y \mid X)] = \mathbb{E}_{X,Y}[S] = \mathbb{E}[S] = \mathbb{E}_S[S] \tag{42}$$

$$= \int p(S = q)\, q\, dq \tag{43}$$

$$\overset{(*)}{=} \int p(T = q)\, q\, dq \tag{44}$$

$$= \mathbb{E}_T[T] = \mathbb{E}[T] = \mathbb{E}_{X,\hat{Y}}[T] = \mathbb{E}[p(\hat{Y} \mid X)], \tag{45}$$

which concludes the proof. $\qquad\square$

The reader is invited to compare this derivation to the corresponding longer proof in the appendix of Jiang et al. (2022). The fully probabilistic perspective greatly simplifies the results, and the proofs are straightforward.

**CACE.** Showing that CACE bounds the gap between test error and disagreement is also straightforward:

**Theorem 4.5.** *For any model* $p(\hat{y} \mid x)$*, we have:*

$$|\mathbb{E}[\text{TestError}] - \mathbb{E}[\text{Dis}]| \leq \text{CACE}.$$

*Proof.* First, we note that

$$\text{CACE} = \int_{q \in [0,1]} \big| p(p(\hat{Y} = Y \mid X) = q) - p(p(\hat{Y} \mid X) = q) \big| dq. \tag{46}$$

following the equivalences in the proof of Lemma 4.2. Then using the triangle inequality for integrals and $0 \leq q \leq 1$, we obtain:

$$\text{CACE} \tag{47}$$

$$= \int_{q \in [0,1]} \big| p(p(\hat{Y} = Y \mid X) = q) - p(p(\hat{Y} = \hat{Y} \mid X) = q) \big| dq \tag{48}$$

$$\geq \int_{q \in [0,1]} q \big| p(p(\hat{Y} = Y \mid X) = q) - p(p(\hat{Y} = \hat{Y} \mid X) = q) \big| dq \tag{49}$$

$$\geq \Big| \int_{q \in [0,1]} q \, p(p(\hat{Y} = Y \mid X) = q) \, dq - \int_{q \in [0,1]} q \, p(p(\hat{Y} \mid X) = q) \, dq \Big|. \tag{50}$$

$$= \big| \mathbb{E}[S] - \mathbb{E}[T] \big| \tag{51}$$

$$= \big| \mathbb{E}[\text{TestError}] - \mathbb{E}[\text{Dis}] \big|, \tag{52}$$

where we have used the monotonicity of integration in (49) and the triangle inequality in (50). □

The bound also serves as another—even simpler—proof for Theorem 4.4:

> *Conclusion* 2. When the Bayesian model satifies class-wise or class-aggregated calibration, we have CACE = 0 and thus $\mathbb{E}[\text{TestError}] = \mathbb{E}[\text{Dis}]$, i.e. the model satisfies GDE.

Furthermore, note again that a Bayesian model was not necessary for the last two theorems. The model parameters $\Omega$ were not mentioned—except for the specific definitions of TestError and Dis which depend on $\Omega$ following Jiang et al. (2022) but which we only use in expectation.

Moreover, we see that we can easily upper-bound CACE using the triangle inequality by 2, narrowing the statement in Jiang et al. (2022) that CACE can lie anywhere in $[0, K]$:

*Conclusion* 3. CACE $\leq 2$.

Additionally, for completeness, we can also define the class-wise calibration error formally and show that it is bounded by CACE using the triangle inequality:

**Definition 4.6.** The *class-wise calibration error (CWCE)* is defined as:

$$\text{CWCE} := \sum_k \int_{q \in [0,1]} \big| p(Y = k, p(\hat{Y} = k \mid X) = q) - p(\hat{Y} = k, p(\hat{Y} = k \mid X) = q) \big|. \tag{53}$$

**Lemma 4.7.** CWCE $\geq$ CACE $\geq |\mathbb{E}[\text{Acc}] - \mathbb{E}[\text{PredAcc}]|$.

Note that when we compute CACE empirically, we divide the dataset into several bins for different intervals of $p(\hat{Y} = k \mid X)$. Jiang et al. (2022) use 15 bins. If we were to use a single bin, we would compute $|\mathbb{E}[\text{Acc}] - \mathbb{E}[\text{PredAcc}]|$ directly.

> In §B we show that CWCE has previously been introduced as 'adaptive calibration error' in Nixon et al. (2019) and CACE as 'static calibration error' (with noteworthy differences between Nixon et al. (2019) and its implementation).

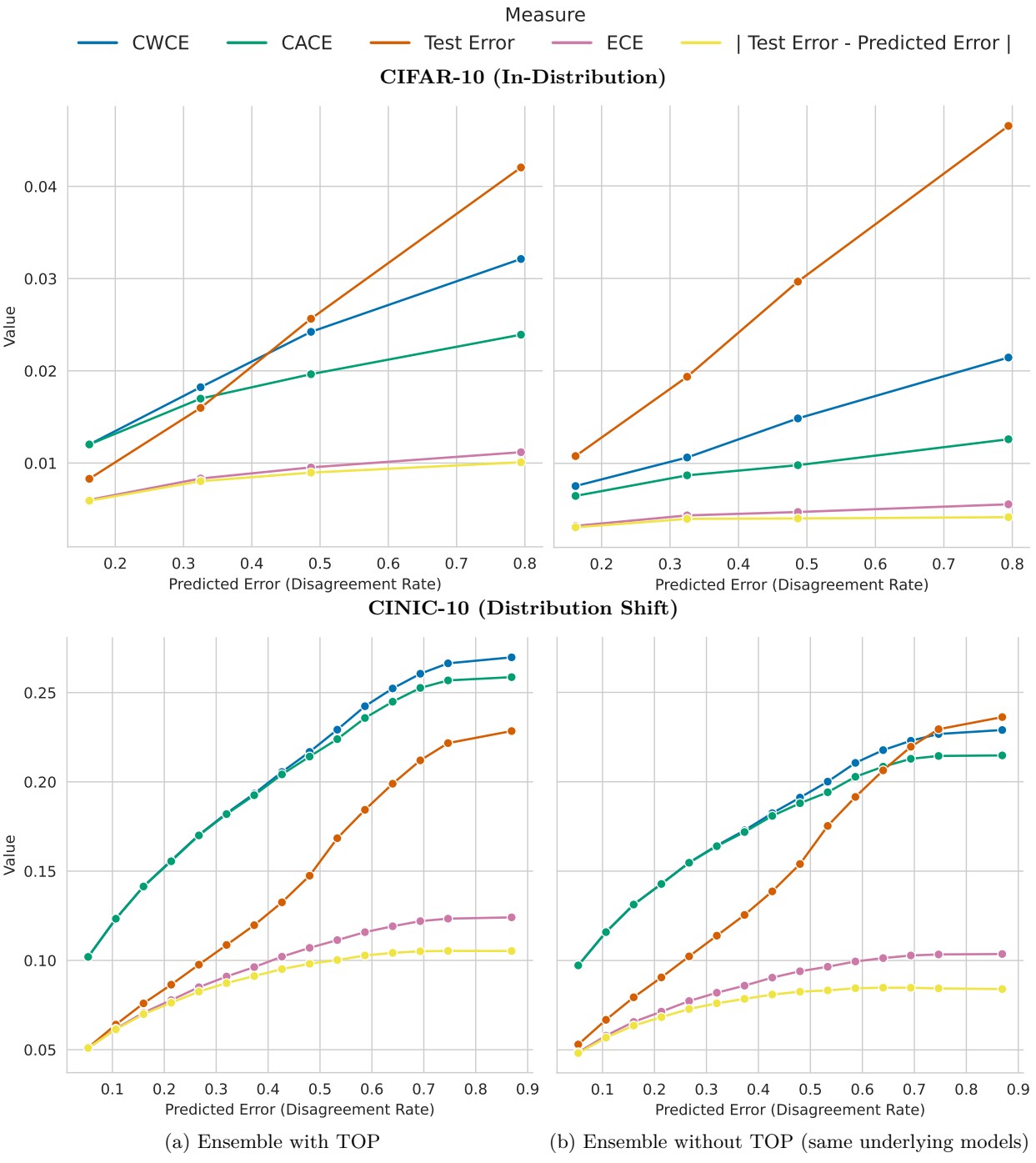

Figure 1: *Rejection Plot of Calibration Metrics for Increasing Disagreement In-Distribution (CIFAR-10) and Under Distribution Shift (CINIC-10).* Different calibration metrics ($ECE$, CWCE, CACE) vary across CIFAR-10 and CINIC-10 on an ensemble of 25 Wide-ResNet-28-10 model trained on CIFAR-10, depending on the rejection threshold of the predicted error (disagreement rate). Thus, calibration cannot be assumed constant for in-distribution data or under distribution shift. The test error increases almost linearly with the predicted error (disagreement rate), leading to 'GDE gap' |Test Error − Predicted Error| becoming almost flat, providing evidence for the empirical observations in Nakkiran & Bansal (2020); Jiang et al. (2022). The mean predicted error (disagreement rate) is shown on the x-axis. **(a)** shows results for an ensemble using TOP (following Jiang et al. (2022)), and **(b)** for a regular deep ensemble without TOP. The regular deep ensemble is better calibrated but has higher test error overall and lower test error for samples with small predicted error.

# 5    Deterioration of Calibration under Increasing Disagreement

Generally, we can only hope to trust model calibration for in-distribution data, while under distribution shift, the calibration ought to deteriorate. In our empirical falsification using models trained on CIFAR-10 and evaluated on the test sets of CIFAR-10 and CINIC-10, as a dataset with a distribution shift, we find in both cases that calibration deteriorates under increasing disagreement. We further examine ImageNET and PACS in §E.2. Most importantly though, calibration markedly worsens under distribution shift.

Specifically, we examine an ensemble of 25 WideResNet models (Zagoruyko & Komodakis, 2016) trained on CIFAR-10 (Krizhevsky et al., 2009) and evaluated on CIFAR-10 and CINIC-10 test data. CINIC-10 (Darlow et al., 2018) consists of CIFAR-10 and downscaled ImageNet samples for the same classes, and thus includes a distribution shift. The training setup follows the one described in Mukhoti et al. (2021), see appendix §E.1.

Figure 1 shows rejection plots under increasing disagreement for in-distribution data (CIFAR-10) and under distribution shift (CINIC-10). The rejection plots threshold the dataset on increasing levels of the predicted error (disagreement rate)—which is a measure of epistemic uncertainty when there is no expected aleatoric uncertainty in the dataset. We examine ECE, class-aggregated calibration error (CACE), class-wise calibration error (CWCE), error $\mathbb{E}[\text{TestError}]$, and 'GDE gap', $|\mathbb{E}[\text{Acc}]-\mathbb{E}[\text{PredAcc}]|$, as the predicted error (disagreement rate), $\mathbb{E}[\text{Dis}] = 1 - \mathbb{E}[\text{PredAcc}]$, increases. We observe that all calibration metrics, ECE, CACE and CWCE, deteriorate under increasing disagreement, both in distribution and under distribution shift, and also worsen under distribution shift overall.

We also observe the same for ImageNet (Deng et al., 2009) and PACS (Li et al., 2017), which we show in appendix §E.2.

This is consistent with the experimental results of Ovadia et al. (2019) which examines dataset shifts. However, given that the calibration metrics change with the quantity of interest, we conclude that:

> *Conclusion* 4. The bound from Theorem 3.6 might not have as much expressive power as hoped since the calibration metrics themselves deteriorate as the model becomes more 'uncertain' about the data.

At the same time, the 'GDE gap', which is the actual gap between test error and predicted error, flattens, and the test error develops an almost linear relationship with the predicted error (up to a bias). This shows that there seem to be intriguing empirical properties of deep ensemble as observed previously (Nakkiran & Bansal, 2020; Jiang et al., 2022). However, they are not explained by the proposed calibration metrics[7].

As described previously, the results are not limited to Bayesian or version-space models but also apply to any model $\text{p}(\hat{y} \,|\, x)$, including a regular deep ensembles without TOP. In our experiment, we find that a regular deep ensemble is better calibrated than the same ensemble made to satisfy TOP. We hypothesize that each ensemble member's own predictive distribution is better calibrated than its one-hot outputs, yielding a better calibrated ensemble overall.

Given that all these calibration metrics require access to the labels, and we cannot assume the model to be calibrated under distribution shift, we might just as well use the labels directly to asses the test error.

# 6    Discussion

Here, we discuss connections to Bayesian model disagreement and epistemic uncertainty, as well as connections to information theory. We expand on these points in much greater detail in appendix §C.

**Bayesian Model Disagreement.** From a Bayesian perspective, as the epistemic uncertainty increases, we expect the model to become less reliable in its predictions. The predicted error of the model is a measure of the model's overall uncertainty, which is the total of aleatoric and epistemic uncertainty and thus correlated with epistemic uncertainty. Thus, we can hypothesize that as the predicted error increases, the model should

---

[7]The simplest explanation is that very few samples have high predicted error and thus the rejection plots flatten. This is not true. For CINIC-10, the first bucket contains 50k samples, and each latter buckets adds additional ∼10k samples.

become less reliable, which will be reflected in increasing calibration metrics. This is exactly what we have empirically validated in the previous section.

**Connection to Information Theory.** At first sight, Jiang et al. (2022) seems disconnected from information theory. However, we can draw a connection by using $\hat{h}(p) := 1 - p$ as a linear approximation for Shannon's information content $h(p) = -\log p$. Semantically, both this approximation and Shannon's information content quantify surprise (i.e., prediction error). Both are 0 for certain events. For unlikely events, the former tends to 1 while the latter tends to $+\infty$.

This leads to common-sense definitions and statements from an information-theoretic point of view. We can even formulate parallel statements using information theory and see that the statements relate to total uncertainty and not epistemic uncertainty in a Bayesian sense.

**Other Related Literature.** Beyond Jiang et al. (2022), this note offers a perspective on Granese et al. (2021) and Garg et al. (2022), which are proposing related approaches.

Granese et al. (2021) use the predicted error (disagreement rate), referred to as $D_\alpha$, and the predicted top-1 error, $D_\beta$, to estimate when the model will be wrong. As noted in §C, the predicted error can be seen as an approximation of Shannon's entropy. Thus, $D_\alpha$ is effectively using an approximation of the prediction entropy for OOD detection and rejection classification. Similarly, $D_\beta$ is the maximum class confidence. Both are well-known baselines for OOD detection (Hendrycks & Gimpel, 2016). There is no ablation to see how $D_\alpha$ and $D_\beta$ differ from these baselines. We leave this to future work. The paper frames the question of whether a model's predictions will be correct as binary classification problem on top of the underlying model's output probabilities and investigate this from a theoretical point of view. They also examine using input perturbations similar to Liang et al. (2017) and Lee et al. (2018).

Garg et al. (2022) threshold the predictive entropy or maximum class confidence to estimate the test error under distribution shift. They estimate the threshold by calibrating it on in-distribution labelled data: the threshold is chosen such that the percentage of rejected in-distribution validation data approximately equals the test error on this in-distribution data. They call this approach *Average Thresholded Confidence (ATC)*. They find that ATC using entropy performs better than ATC using the maximum confidence and other approaches, including GDE. Their results show that ATC also degrades under increasing distribution shifts similar to what we have seen for GDE in §5 as the choice of threshold is explicitly tied to the in-distribution[8]. Garg et al. (2022) explicitly examine the theoretical limits when no further assumptions are made.

## 7 Conclusion

We have found that the theoretical statements in Jiang et al. (2022) can be expressed and proven more concisely when using probabilistic notation for (Bayesian) models that output softmax probabilities.

Moreover, we empirically found the proposed calibration metrics to deteriorate under increasing disagreement for in-distribution data, and as expected, we have found the same behavior under distribution shifts.

While Jiang et al. (2022) are careful to qualify their results for distribution shifts, above results should give us pause: strong assumptions are still needed to conjecture about model generalization, and we need to beware of circular arguments.

## Acknowledgements

The authors would like to thank the anonymous TMLR reviewers for their kind, constructive, and helpful feedback during the review process. Moreover, we would like to thank Jannik Kossen, Joost van Amersfoort, and the members of OATML in general for their feedback at various stages of the project, and the authors for Jiang et al. (2022) for constructively engaging with a preprint of this work. AK is supported by the UK EPSRC CDT in Autonomous Intelligent Machines and Systems (grant reference EP/L015897/1).

---

[8]See also Figures 7, 8, and 9 in the appendix of Garg et al. (2022)

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

# A  Equivalent Definitions

Jiang et al. (2022) defines a *hypothesis space* $\mathcal{H}$. In the literature, this is also sometimes called a version space. The hypothesis space induced by a stochastic training algorithm $\mathcal{A}$ is named $\mathcal{H}_{\mathcal{A}}$.

We can identify each hypothesis $h : \mathcal{X} \to [K]$ with itself as parameter $\omega_h = h$ and define $p(\omega_h)$ as a uniform distribution over all parameters/hypotheses in $\mathcal{H}_{\mathcal{A}}$. This has the advantage of formalizing the distribution from which hypothesis are drawn ($h \sim \mathcal{H}_{\mathcal{A}}$), which is not made explicit in Jiang et al. (2022). $h(x) = k$ then becomes $\arg\max_{\hat{y}} p(\hat{y} \mid x, \omega_h) = k$. Moreover, as $p(\hat{y}, \omega \mid x)$ satisfies TOP, we have[9]

$$``\mathbb{1}\{h(x) = k\}'' = p(\hat{Y} = k \mid x, \omega_h). \tag{54}$$

Thus, "$\mathrm{TestErr}_{\mathscr{D}}(h) \triangleq \mathbb{E}_{\mathscr{D}}[\mathbb{1}[h(X) \neq Y]]$" is equivalent to:

$$``\,\mathrm{TestErr}_{\mathscr{D}}(h) = \mathbb{E}_{\mathscr{D}}[\mathbb{1}[h(X) \neq Y]]'' \tag{55}$$
$$= \mathbb{E}_{X,Y}[p(\hat{Y} \neq Y \mid X, \omega_h)] \tag{56}$$
$$= \mathbb{E}_{X}[\mathbb{P}[\hat{Y} \neq Y \mid X, \omega_h]] \tag{57}$$
$$= \mathbb{P}[\hat{Y} \neq Y \mid \omega_h] \tag{58}$$
$$= \mathrm{TestError}(\omega_h). \tag{59}$$

Similarly, "$\mathrm{Dis}_{\mathscr{D}}(h, h') \triangleq \mathbb{E}_{\mathscr{D}}[\mathbb{1}[h(X) \neq h'(X)]]$" is equivalent to:

$$``\,\mathrm{Dis}_{\mathscr{D}}(h, h') \triangleq \mathbb{E}_{\mathscr{D}}[\mathbb{1}[h(X) \neq h'(X)]]'' \tag{60}$$
$$= \mathbb{E}_{\hat{Y}, \hat{Y}'}[\mathbb{P}[\hat{Y} \neq \hat{Y}' \mid \omega_h, \omega_{h'}]] \tag{61}$$
$$= \mathrm{Dis}(\omega_h, \omega_{h'}). \tag{62}$$

Further, "$\tilde{h}_k(x) \triangleq \mathbb{E}_{\mathscr{H}_{\mathcal{A}}}[\mathbb{1}[h(x) = k]]$" is equivalent to:

$$``\tilde{h}_k(x) \triangleq \mathbb{E}_{\mathscr{H}_{\mathcal{A}}}[\mathbb{1}[h(x) = k]]'' \tag{63}$$
$$= \mathbb{E}_{\Omega}[p(\hat{Y} = k \mid x, \Omega)] \tag{64}$$
$$= p(\hat{Y} = k \mid x). \tag{65}$$

For the GDE, "$\mathbb{E}_{h,h' \sim \mathscr{H}_{\mathcal{A}}}[\mathrm{Dis}_{\mathscr{D}}(h, h')] = \mathbb{E}_{h \sim \mathscr{H}_{\mathcal{A}}}[\mathrm{TestErr}(h)]$" is equivalent to:

$$``\mathbb{E}_{h,h' \sim \mathscr{H}_{\mathcal{A}}}[\mathrm{Dis}_{\mathscr{D}}(h, h')] = \mathbb{E}_{h \sim \mathscr{H}_{\mathcal{A}}}[\mathrm{TestErr}(h)]''$$
$$\Leftrightarrow \mathbb{E}_{\Omega, \Omega'}[\mathrm{Dis}(\Omega, \Omega')] = \mathbb{E}_{\Omega}[\mathrm{TestError}(\Omega)]. \tag{66}$$

For the class-wise calibration, "$p(Y = k \mid \tilde{h}_k(X) = q) = q$" is equivalent to:

$$``p(Y = k \mid \tilde{h}_k(X) = q) = q'' \tag{67}$$
$$\Leftrightarrow p(Y = k \mid p(\hat{Y} = k \mid X) = q) = q. \tag{68}$$

For the class-aggregated calibration, "$\frac{\sum_{k=0}^{K-1} p(Y=k, \tilde{h}_k(X)=q)}{\sum_{k=0}^{K-1} p(\tilde{h}_k(X)=q)} = q$" (and note in Jiang et al. (2022), class indices run from $0..K-1$) is equivalent to:

$$``\frac{\sum_{k=0}^{K-1} p(Y = k, \tilde{h}_k(X) = q)}{\sum_{k=0}^{K-1} p(\tilde{h}_k(X) = q)} = q'' \tag{69}$$

---

[9] We put definitions and expressions written using the notation and variables from Jiang et al. (2022) inside quotation marks "" to avoid ambiguities.

$$\Leftrightarrow \frac{\sum_{k=1}^{K} \mathrm{p}(Y = k, \mathrm{p}(\hat{Y} = k \mid X) = q)}{\sum_{k=1}^{K} \mathrm{p}(\mathrm{p}(\hat{Y} = k \mid X) = q)} = q \tag{70}$$

$$\Leftrightarrow \sum_{k=1}^{K} \mathrm{p}(Y = k, \mathrm{p}(\hat{Y} = k \mid X) = q)$$

$$= q \sum_{k=1}^{K} \mathrm{p}(\mathrm{p}(\hat{Y} = k \mid X) = q). \tag{71}$$

Finally, for the class-aggregated calibration error, the definition is equivalent to:

$$\text{``} \mathrm{CACE}_{\mathscr{D}}(\tilde{h})$$

$$\triangleq \int_{q \in [0,1]} \left| \frac{\sum_{k} p\left(Y = k, \tilde{h}_k(X) = q\right)}{\sum_{k} p\left(\tilde{h}_k(X) = q\right)} - q \right| \cdot \sum_{k} p\left(\tilde{h}_k(X) = q\right) dq \tag{72}$$

$$= \int_{q \in [0,1]} \left| \sum_{k} p\left(Y = k, \tilde{h}_k(X) = q\right) - q \sum_{k} p\left(\tilde{h}_k(X) = q\right) \right| dq'' \tag{73}$$

$$= \int_{q \in [0,1]} \left| \sum_{k} \mathrm{p}(Y = k, \mathrm{p}(\hat{Y} = k \mid X) = q) - q \sum_{k} \mathrm{p}(\mathrm{p}(\hat{Y} = k \mid X) = q) \right| dq \tag{74}$$

# B   Comparison of CACE and CWCE with calibration metrics with 'adaptive calibration error' and 'static calibration error'

Nixon et al. (2019) examine shortcomings of the ECE metric and identify a lack of class conditionality, adaptivity and the focus on the maximum probability (argmax class) as issues. They suggest an adaptive calibration error which uses adaptive binning and averages of the calibration error separately for each class, thus equivalent to the class-wise calibration error and class-wise calibration (up to adaptive vs. even binning). In the paper, the static calibration error is defined as ACE with even instead of adaptive binning. However, in the widely used implementation[10], SCE is defined as equivalent to the class-aggregated calibration error.

# C   Expanded Discussion

Here, we discuss connections to Bayesian model disagreement and epistemic uncertainty, as well as connections to information theory, the bias-variance trade-off, and prior literature.

## C.1   Bayesian Model Disagreement

From a Bayesian perspective, as the epistemic uncertainty increases, we expect the model to become less reliable in its predictions. The predicted error of the model is a measure of the overall uncertainty of the model which is the total of aleatoric and epistemic uncertainty, and thus correlated with epistemic uncertainty. Thus, we can hypothesize that as the predicted error increases, the model should become less reliable, which will be reflected in increasing calibration metrics. This is what we have empirically validated in the previous section. We can expand on the connection to the Bayesian perspective. In particular, we can connect the statements of Jiang et al. (2022) to a well-known Bayesian measure of model disagreement.

**Information Theory.** We follow notation introduced in Kirsch & Gal (2021). In particular, for entropy, we use an implicit or explicit notation, $\mathrm{H}[X]$ or $\mathrm{H}(\mathrm{p}(X))$, and use a notation for the cross-entropy $\mathrm{H}(\mathrm{p} \parallel \mathrm{q})$ which is follows the Kullback-Leibler divergence $\mathrm{D}_{\mathrm{KL}}(\mathrm{p} \parallel \mathrm{q})$. We base these definitions on Shannon's information content $\mathrm{h}(p)$:

$$\mathrm{h}(p) := -\log p, \tag{75}$$

---

[10]https://github.com/google-research/robustness_metrics/blob/baa47fbe38f80913590545fe7c199898f9aff349/robustness_metrics/metrics/uncertainty.py#L1585, added in April 2021

$$H(p(X) \parallel q(X)) := \mathbb{E}_{p(x)}[h(q(x))], \tag{76}$$

$$H[X] := H(p(X)) := H(p(X) \parallel p(X)), \tag{77}$$

$$D_{KL}(p(X) \parallel q(X)) := H(p(X) \parallel q(X)) - H(p(X)), \tag{78}$$

where p and q are probabilities distributions. Conditional and joint entropies are defined as usual. For completeness, for random variables $X$ and $Y$:

$$H[X \mid Y] := \mathbb{E}_{p(x,y)}[-\log p(x \mid y)]. \tag{79}$$

Finally, the mutual information $I[X;Y]$ for random variables $X$, $Y$ is defined as expected uncertainty reduction—also sometimes called expected information gain (Lindley, 1956):

$$I[X;Y] = H[X] - H[X \mid Y], \tag{80}$$

as the entropy $H[X]$ can be seen as quantifying the 'uncertainty' about the random variable $X$, and $H[X \mid Y]$ as expected uncertainty about $X$ after observing $Y$.

**Bayesian Model Disagreement.** The mutual information $I[\hat{Y}; \Omega \mid x]$ between predictions $\hat{Y}$ and model parameters $\Omega$ for a given sample $x$ is a well-known quantity that measures the model disagreement for that sample. It can be computed without having to perform Bayesian inference:

$$I[\hat{Y}; \Omega \mid x] = H[\hat{Y} \mid x] - H[\hat{Y} \mid x, \Omega] \tag{81}$$

$$= \mathbb{E}_{\hat{Y}, X} \big[ -\log p(\hat{Y} \mid X) \tag{82}$$

$$- \mathbb{E}_{\Omega}[-\log p(\hat{Y} \mid X, \Omega)] \big]$$

To see that $I[\hat{Y}; \Omega \mid x]$ measures model disagreement, observe that if each $\omega$ was equally likely to obtain a different one-hot class prediction $\hat{Y}$, the second term would be 0, while the first term $H[\hat{Y} \mid x]$ would be maximal $= \log K$ as the predictive distribution would be uniform. Equally, if the predictions for each $\omega$ agreed, we would have $p(\hat{y} \mid x) = p(\hat{y} \mid x, \omega)$ for all $\omega$, the first and second term would be equal, and the mutual information would be 0 (Kirsch et al., 2019).

Model disagreement is a proxy of epistemic uncertainty. Epistemic uncertainty measures model-dependent uncertainty (i.e. reliability) and is also known as *reducible* uncertainty: if we were to update the model with a label for $x$ using Bayesian inference, the updated model's uncertainty for $x$ would reduce. For this reason, $I[\hat{Y}; \Omega \mid x]$ is often used in Bayesian active learning, where it is known as *BALD (Bayesian Active Learning by Disagreement)* (Houlsby et al., 2011), or in Bayesian optimal experiment design, where it is known as *Expected Information Gain* (Lindley, 1956). In non-Bayesian active learning, the very same term is known as *Query-by-Committee* (McCallum & Nigam, 1998) and computed via a Kullback-Leilber divergence:

$$I[\hat{Y}; \Omega \mid x] = H[\hat{Y} \mid x] - H[\hat{Y} \mid x, \Omega]$$

$$= D_{KL}(p(\hat{Y} \mid x, \Omega) \parallel p(\hat{Y} \mid x)). \tag{83}$$

Taking an expectation, we obtain the expected model disagreement over the data $I[\hat{Y}; \Omega \mid X] = \mathbb{E}_X[I[\hat{Y}; \Omega \mid X]]$.

In §E.2, we also report empirical results for rejection plots based on Bayesian model disagreement instead of predicted error.

### C.2 Connection to Information Theory

At first sight, Jiang et al. (2022) seems disconnected from information theory. However, we can recover statements by using $\hat{h}(p) := 1 - p$ as a linear approximation for Shannon's information content $h(p)$:

$$\hat{h}(p) = 1 - p \le -\log p = h(p). \tag{84}$$

$\hat{h}(p)$ is just the first-order Taylor expansion of $h(p) = -\log p$ around 1. Semantically, both Shannon's information content and this approximation quantify surprise. Both are 0 for certain events. For unlikely events, the former tends to $+\infty$ while the latter tends to 1.

We can define an *approximate entropy* $\hat{\mathrm{H}}[X]$ using h$'$:

$$\hat{\mathrm{H}}[X] := \mathbb{E}[\hat{\mathrm{h}}(\mathrm{p}(X))] = 1 - \mathbb{E}[\mathrm{p}(x)] = 1 - \sum_x \mathrm{p}(x)^2, \tag{85}$$

and an *approximate mutual information* $\hat{\mathrm{I}}[X;Y]$:

$$\hat{\mathrm{I}}[X;Y] := \hat{\mathrm{H}}[X] - \hat{\mathrm{H}}[X \mid Y] = \hat{\mathrm{H}}[X] - \mathbb{E}_{\mathrm{p}(y)} \hat{\mathrm{H}}[X \mid y], \tag{86}$$

following the semantic notion of mutual information as expected information gain in §2.

$\hat{\mathrm{I}}[\hat{Y}; \Omega \mid x]$ **as Covariance Trace.** This approximate mutual information has a surprisingly nice property, which was detailed in Smith & Gal (2018) originally:

**Proposition C.1.** *The approximate mutual information $\hat{\mathrm{I}}[\hat{Y}; \Omega \mid x]$ is equal the sum of the variances of $\hat{y} \mid x, \Omega$ over all $\hat{y}$:*

$$\hat{\mathrm{I}}[\hat{Y}; \Omega \mid x] = \sum_{\hat{y}=1}^{K} \mathrm{Var}_\Omega[\mathrm{p}(\hat{y} \mid x, \Omega)] \geq 0. \tag{87}$$

We present a proof in §D in the appendix. The sum of variances of the predictive probabilities (or trace of the respective covariance matrix) is a common proxy for epistemic uncertainty (Gal et al., 2017), and here the mutual information $\hat{\mathrm{I}}[\hat{Y}; \Omega \mid x]$ using $\hat{\mathrm{h}}$ is just that. This gives evidence that these definitions are sensible and connects them to other prior Bayesian literature. Importantly, this also shows that $\hat{\mathrm{H}}[\hat{Y} \mid x] \geq \hat{\mathrm{H}}[\hat{Y} \mid x, \Omega]$.

**Connection to Jiang et al. (2022).** As random variable of $X$ and $Y$, $\hat{\mathrm{H}}[\hat{Y} = Y \mid X]$ is the test error:

$$\hat{\mathrm{H}}[\hat{Y} = Y \mid X] = 1 - \mathrm{p}(\hat{Y} = Y \mid X) = \mathrm{TestError}. \tag{88}$$

Thus, the approximate cross-entropy

$$\hat{\mathrm{H}}(\mathrm{p}(Y \mid X) \| \mathrm{p}(\hat{Y} = Y \mid X)) = \mathbb{E}_{\mathrm{p}(Y \mid X)}[\hat{\mathrm{h}}(\mathrm{p}(\hat{Y} = Y \mid X))] \tag{89}$$

is the expected test error $\mathbb{E}[\mathrm{TestError}]$.

Similarly, when TOP is fulfilled, the mutual information $\hat{\mathrm{I}}[\hat{Y}; \Omega \mid X]$ is the expected disagreement rate $\mathbb{E}[\mathrm{Dis}]$. That is, when $\hat{Y} \mid X, \Omega$ is one-hot, we have:

$$\hat{\mathrm{H}}[\hat{Y} \mid X, \Omega] = 1 - \mathbb{E}_X \underbrace{\mathbb{E}_{\hat{Y}}[\mathrm{p}(\hat{Y} \mid X, \Omega) \mid X]}_{=1} = 0. \tag{90}$$

and thus:

$$\hat{\mathrm{I}}[\hat{Y}; \Omega \mid X] = \hat{\mathrm{H}}[\hat{Y} \mid X] - \hat{\mathrm{H}}[\hat{Y} \mid X, \Omega] \tag{91}$$

$$= \hat{\mathrm{H}}[\hat{Y} \mid X] \tag{92}$$

$$= 1 - \mathbb{E}_{X,\hat{Y}}[\mathrm{p}(\hat{Y} \mid X)] \tag{93}$$

$$= \mathbb{E}[\mathrm{Dis}]. \tag{94}$$

**Lemma C.2.** *When the model $\mathrm{p}(\hat{y} \mid x, \omega)$ satisfies TOP, the GDE is equivalent to:*

$$\hat{\mathrm{H}}(\mathrm{p}(Y \mid X) \| \mathrm{p}(\hat{Y} = Y \mid X)) = \hat{\mathrm{I}}[\hat{Y}; \Omega \mid X]. \tag{95}$$

This relates the approximate cross-entropy loss (test error) to the approximate Bayesian model disagreement.

**Without TOP.** If TOP does not hold, the *actual* expected disagreement $\hat{\mathrm{I}}[\hat{Y}; \Omega \mid x]$ lower-bounds the "expected disagreement rate" $\mathbb{E}[\mathrm{Dis}]$, which then equals the expected *predicted* error $1 - \mathbb{E}_{X,\hat{Y}}[\mathrm{p}(\hat{Y} \mid X)]$ when we have GDE. We have the following general equivalence to GDE:

**Lemma C.3.** *For a model* $p(\hat{y} \mid x)$*, the GDE is equivalent to:*

$$\hat{H}(p(Y \mid X) \parallel p(\hat{Y} = Y \mid X)) = \hat{H}[\hat{Y} \mid X] \geq \hat{I}[\hat{Y}; \Omega \mid X]. \tag{96}$$

The other statements and proofs translate likewise, and intuitively seem sensible from an information-theoretic perspective. We can go further and directly establish analogous properties using information theory in the next subsection.

### C.3 Information-Theoretic Version

Here, we derive an information-theoretic version of the GDE both under the assumption of TOP and without. Importantly, we will not require a Bayesian model for any of the main statements as they hold for any model $p(\hat{y} \mid x)$. We show that we can artificially introduce a connection to disagreement using TOP.

**Information-Theoretic GDE.** We have already introduced the BALD equation eq. (81), which connects expected disagreement and predictive uncertainty:

$$I[\hat{Y}; \Omega \mid x] = H[\hat{Y} \mid x] - H[\hat{Y} \mid x, \Omega]$$

The expected disagreement is measured by the mutual information $I[\hat{Y}; \Omega \mid X]$, and the prediction error is measured by the cross-entropy of the predictive distribution under the true data generating distribution $H(p(Y \mid X) \parallel p(\hat{Y} = Y \mid X))$. Indeed, the test error is bounded by it (Kirsch et al., 2020):

$$p(Y \neq \hat{Y}) \leq 1 - e^{-H(p(Y|X)\parallel p(\hat{Y}=Y|X))}. \tag{97}$$

When our model fulfills TOP, we have $H[\hat{Y} \mid X, \Omega] = 0$, and thus $I[\hat{Y}; \Omega \mid X] = H[\hat{Y} \mid X]$. The expected disagreement then equals the predicted label uncertainty $H[\hat{Y} \mid X]$. Generally, we can define an 'entropic GDE':

**Definition C.4.** A model $p(\hat{y} \mid x)$ satifies entropic GDE, when:

$$H(p(Y \mid X) \parallel p(\hat{Y} = Y \mid X)) = H[\hat{Y} \mid X]. \tag{98}$$

**Lemma C.5.** *When a Bayesian model* $p(\hat{y}, \omega \mid x)$ *satisfies TOP, entropic GDE is equivalent to*

$$H(p(Y \mid X) \parallel p(\hat{Y} = Y \mid X)) = I[\hat{Y}; \Omega \mid X]. \tag{99}$$

The latter is close to GDE, especially when comparing to the previous section.

We can formulate an entropic class-aggregated calibration by connecting $H[\hat{y} \mid x]$ with $H[y \mid x]$, where we define $H[\hat{y} \mid x] := h(p(\hat{y} \mid x)) = -\log p(\hat{y} \mid x)$ (Kirsch & Gal, 2021). That is, instead of using probabilities, we use Shannon's information-content:

**Definition C.6.** The model $p(\hat{y} \mid x)$ satisfies *entropic class-aggregated calibration* when for any $q \geq 0$:

$$p(H[\hat{Y} = Y \mid X] = q) = p(H[\hat{Y} = \hat{Y} \mid X] = q). \tag{100}$$

Similarly, we can define the *entropic class-aggregated calibration error (ECACE)*:

$$\text{ECACE} := \int_{q \in [0,\infty)} \big| p(H[\hat{Y} = Y \mid X] = q) \tag{101}$$
$$- p(H[\hat{Y} = \hat{Y} \mid X] = q) \big| dq.$$

As $-\log p$ is strictly monotonic and thus invertible for non-negative $p$, entropic class-aggregated calibration and class-aggregated calibration are equivalent. ECACE and CACE are not, though.

The expectation of the transformed random variable $H[\hat{Y} = Y \mid X]$ (in $Y$ and $X$) is just the cross-entropy:

$$\mathbb{E}_{X,Y} H[\hat{Y} = Y \mid X] = \mathbb{E}_{p(x,y)} H[\hat{Y} = y \mid X] = H(p(Y \mid X) \parallel p(\hat{Y} = Y \mid X)). \tag{102}$$

Using this notation, and analogous to Theorem 3.6, we can show:

**Theorem C.7.** *When* $\mathrm{H}[\hat{y} \mid x] = -\log \mathrm{p}(\hat{y} \mid x)$ *is uppper-bounded by* $L$ *for all* $\hat{y}$ *and* $x$, *we have:*

$$\text{ECACE} \geq \frac{1}{L}\big|\mathrm{H}(\mathrm{p}(Y \mid X) \,\|\, \mathrm{p}(\hat{Y} = Y \mid X)) - \mathrm{H}[\hat{Y} \mid X]\big|, \tag{103}$$

*and when the model satisfies TOP, equivalently:*

$$= \frac{1}{L}\big|\mathrm{H}(\mathrm{p}(Y \mid X) \,\|\, \mathrm{p}(\hat{Y} = Y \mid X)) - \mathrm{I}[\hat{Y}; \Omega \mid X]\big|. \tag{104}$$

There might be better conditions than the upper-bound above but this bound is in the spirit of Jiang et al. (2022). Indeed, the proof of Theorem 3.6 is the same, except that we use $q \leq L$ instead of $q \leq 1$. Finally, when the model satisfies entropic class-aggregated calibration, ECACE $= 0$, cross-entropy (or negative expected log likelihood) equals disagreement (respectively, predicted label uncertainty when TOP does not hold). Thus, we have:

**Theorem C.8.** *When the model* $\mathrm{p}(\hat{y} \mid x)$ *satisfies* entropic class-aggregated calibration, *it trivially also satisfies entropic GDE:*

$$\mathrm{H}(\mathrm{p}(Y \mid X) \,\|\, \mathrm{p}(\hat{Y} = Y \mid X)) = \mathrm{H}[\hat{Y} \mid X] \geq \mathrm{I}[\hat{Y}; \Omega \mid X], \tag{105}$$

*and when TOP holds:*

$$\mathrm{H}(\mathrm{p}(Y \mid X) \,\|\, \mathrm{p}(\hat{Y} = Y \mid X)) = \mathrm{I}[\hat{Y}; \Omega \mid X]. \tag{106}$$

**Without TOP.** Again, if we do not expect one-hot predictions for our ensemble members, the analogy put forward in Jiang et al. (2022) breaks down because the Bayesian disagreement $\mathrm{I}[\hat{Y}; \Omega \mid X]$ only lower bounds the predicted label uncertainty $\mathrm{H}[\hat{Y} \mid X]$ and can not be connected to ECACE the same way. But this also breaks down in the regular version in Jiang et al. (2022).

## D    Additional Proofs

**Lemma D.1.** *For a model* $\mathrm{p}(\hat{y} \mid x)$, *we have for all* $k \in [K]$ *and* $q \in [0, 1]$:

$$\mathrm{p}(\hat{Y} = k \mid \mathrm{p}(\hat{Y} = k \mid X) = q) = q, \tag{107}$$

*when the left-hand side is well-defined.*

*Proof.* This is equivalent to

$$\mathrm{p}(\hat{Y} = k, \mathrm{p}(\hat{Y} = k \mid X) = q) = q\, \mathrm{p}(\mathrm{p}(\hat{Y} = k \mid X) = q), \tag{108}$$

as the conditional probability is either defined or $\mathrm{p}(\mathrm{p}(\hat{Y} = k \mid X) = q) = 0$. Assume the former. Let $\mathrm{p}(\mathrm{p}(\hat{Y} = k \mid X) = q) > 0$. Introducing the auxiliary random variable $T_k := \mathrm{p}(\hat{Y} = k \mid X)$ as a transformed random variable of $X$, we have

$$\mathrm{p}(\hat{Y} = k, T_k = q) = q\, \mathrm{p}(T_k = q). \tag{109}$$

We can write the probability as an expectation over an indicator function

$$\mathrm{p}(\hat{Y} = k, T_k = q) \tag{110}$$

$$= \mathbb{E}_{X,\hat{Y}}[\mathbb{1}\{\hat{Y} = k, T_k(X) = q\}] \tag{111}$$

$$= \mathbb{E}_{X,\hat{Y}}[\mathbb{1}\{\hat{Y} = k\}\,\mathbb{1}\{T_k(X) = q\}] \tag{112}$$

$$= \mathbb{E}_X[\mathbb{1}\{T_k(X) = q\}\,\mathbb{E}_{\hat{Y}}[\mathbb{1}\{\hat{Y} = k\} \mid X]] \tag{113}$$

$$= \mathbb{E}_X[\mathbb{1}\{T_k(X) = q\}\,\mathrm{p}(\hat{Y} = k \mid X)]. \tag{114}$$

Importantly, if $\mathbb{1}\{T_k(x) = q\} = 1$ for an $x$, we have $T_k(x) = \mathrm{p}(\hat{Y} = k \mid x) = q$, and otherwise, we multiply with 0. Thus, this is equivalent to

$$= \mathbb{E}_X[\mathbb{1}\{T_k(X) = q\}\,q] \tag{115}$$
$$= q\,\mathbb{E}_X[\mathbb{1}\{T_k(X) = q\}] \tag{116}$$
$$= q\,\mathrm{p}(T_k(X) = q). \tag{117}$$

$\square$

**Lemma 4.2.** *The model* $\mathrm{p}(\hat{y} \mid x)$ *satisfies* class-wise calibration *when for any* $q \in [0,1]$ *and any class* $k \in [K]$*:*

$$\mathrm{p}(Y = k, \mathrm{p}(\hat{Y} = k \mid X) = q) = \mathrm{p}(\hat{Y} = k, \mathrm{p}(\hat{Y} = k \mid X) = q). \tag{36}$$

*Similarly, the model* $\mathrm{p}(\hat{y} \mid x)$ *satisfies* class-aggregated calibration *when for any* $q \in [0,1]$*:*

$$\mathrm{p}(\mathrm{p}(\hat{Y} = Y \mid X) = q) = \mathrm{p}(\mathrm{p}(\hat{Y} \mid X) = q), \tag{37}$$

*and* class-wise *calibration implies* class-aggregate *calibration.*

*Proof.* Beginning from

$$\mathrm{p}(Y = k \mid \mathrm{p}(\hat{Y} = k \mid X) = q) = q, \tag{118}$$

we expand the conditional probability to

$$\Leftrightarrow \mathrm{p}(Y = k, \mathrm{p}(\hat{Y} = k \mid X) = q) = q\,\mathrm{p}(\mathrm{p}(\hat{Y} = k \mid X) = q), \tag{119}$$

and substitute eq. (35) into the outer $q$, obtaining the first equivalence

$$\Leftrightarrow \mathrm{p}(Y = k, \mathrm{p}(\hat{Y} = k \mid X) = q) = \mathrm{p}(\hat{Y} = k, \mathrm{p}(\hat{Y} = k \mid X) = q). \tag{120}$$

For the second equivalence, we follow the same approach. Beginning from

$$\sum_k \mathrm{p}(Y = k, \mathrm{p}(\hat{Y} = k \mid X) = q) = q\sum_k \mathrm{p}(\mathrm{p}(\hat{Y} = k \mid X) = q), \tag{121}$$

we pull the outer $q$ into the sum and expand using (35)

$$\Leftrightarrow \sum_k \mathrm{p}(Y = k, \mathrm{p}(\hat{Y} = k \mid X) = q) = \sum_k q\,\mathrm{p}(\mathrm{p}(\hat{Y} = k \mid X) = q) = \sum_k \mathrm{p}(\hat{Y} = k, \mathrm{p}(\hat{Y} = k \mid X) = q). \tag{122}$$

In the inner expression, $k$ is tied to $Y$ on the left-hand side and $\hat{Y}$ on the right-hand side, so we have

$$\Leftrightarrow \sum_k \mathrm{p}(Y = k, \mathrm{p}(\hat{Y} = Y \mid X) = q) = \sum_k \mathrm{p}(\hat{Y} = k, \mathrm{p}(\hat{Y} \mid X) = q). \tag{123}$$

Summing over $k$, marginalizes out $Y = k$ and $\hat{Y} = k$ respectively, yielding the second equivalence

$$\Leftrightarrow \mathrm{p}(\mathrm{p}(\hat{Y} = Y \mid X) = q) = \mathrm{p}(\mathrm{p}(\hat{Y} \mid X) = q). \tag{124}$$

Finally, class-wise calibration implies class-aggregated calibration as summing over different $k$ in (120), which is equivalent to class-wise calibration, yields (122), which is equivalent to class-aggregated calibration. $\square$

**Proposition C.1.** *The approximate mutual information* $\hat{\mathrm{I}}[\hat{Y}; \Omega \mid x]$ *is equal the sum of the variances of* $\hat{y} \mid x, \Omega$ *over all* $\hat{y}$*:*

$$\hat{\mathrm{I}}[\hat{Y}; \Omega \mid x] = \sum_{\hat{y}=1}^{K} \mathrm{Var}_\Omega[\mathrm{p}(\hat{y} \mid x, \Omega)] \geq 0. \tag{87}$$

*Proof.* We show that both sides are equal:

$$\hat{I}[\hat{Y}; \Omega \mid x] = \hat{H}[\hat{Y} \mid x] - \hat{H}[\hat{Y} \mid x, \Omega] \tag{125}$$

$$= \mathbb{E}_{\hat{Y}}[1 - p(\hat{Y} \mid x)] - \mathbb{E}_{\hat{Y}, \Omega}[1 - p(\hat{Y} \mid x, \Omega)] \tag{126}$$

$$= \mathbb{E}_{\hat{Y}, \Omega}[p(\hat{Y} \mid x, \Omega)] - \mathbb{E}_{\hat{Y}}[p(\hat{Y} \mid x)] \tag{127}$$

$$= \mathbb{E}_{\Omega} \mathbb{E}_{p(\hat{y}, x, \Omega)}[p(\hat{y} \mid x, \Omega)] - \mathbb{E}_{p(\hat{y} \mid x)} p(\hat{y} \mid x) \tag{128}$$

$$= \mathbb{E}_{\Omega}\left[\sum_{\hat{y}=1}^{K} p(\hat{y} \mid x, \Omega)^2\right] - \sum_{\hat{y}=1}^{K} \mathbb{E}_{\Omega}[p(\hat{y} \mid x, \Omega)]^2 \tag{129}$$

$$= \sum_{\hat{y}=1}^{K} \mathbb{E}_{\Omega}\left[p(\hat{y} \mid x, \Omega)^2\right] - \mathbb{E}_{\Omega}[p(\hat{y} \mid x, \Omega)]^2 \tag{130}$$

$$= \sum_{\hat{y}=1}^{K} \mathrm{Var}_{\Omega}[p(\hat{y} \mid x, \Omega)] \tag{131}$$

$$\geq 0, \tag{132}$$

where we have used that $\mathbb{E}_{p(\hat{y}|x)} p(\hat{y} \mid x) = \sum_{\hat{y}=1}^{K} p(\hat{y} \mid x)^2$. $\square$

# E  Experimental Validation of Calibration Deterioration under Increasing Disagreement

Here, we discuss additional details to allow for reproduction and present results on additional datasets. In addition to the experiments on CIFAR-10 (Krizhevsky et al., 2009) and CINIC-10 (Darlow et al., 2018), we report results for ImageNet (Deng et al., 2009) (in-distribution) using an ensemble of pretrained models and PACS (Li et al., 2017) (distribution shift) where we fine-tune ImageNet models on PACS' 'photo' domain, which is close to ImageNet as source domain, and evaluate it on PACS' 'art painting', 'sketch', and 'cartoon' domains. We use all three domains together for distribution shift evaluation to have more samples for the rejection plots.

## E.1  Experiment Setup

We use PyTorch (Paszke et al., 2019) for all experiments.

**CIFAR-10 and CINIC-10.** We follow the training setup from Mukhoti et al. (2021): we train 25 WideResNet-28-10 models (Zagoruyko & Komodakis, 2016) for 350 epochs on CIFAR-10. We use SGD with a learning rate of 0.1 and momentum of 0.9. We use a learning rate schedule with a decay of 10 at 150 and 250 epochs.

**ImageNet and PACS.** We use pretrained models with various architectures (specifically: ResNet-152-D (He et al., 2018), BEiT-L/16 (Bao et al., 2021), ConvNext-L (Liu et al., 2022), DeiT3-L/16 (Touvron et al., 2020), and ViT-B/16 (Dosovitskiy et al., 2020)) from the timm package (Wightman, 2019) as base models. We freeze all weights except for the final linear layer, which we fine-tune on PACS' 'photo' domain using Adam (Kingma & Ba, 2014) with learning rate $5 \times 10^{-3}$ and batch size 128 for 1000 steps. We then build an ensemble using these different models.

## E.2  Additional Results

In Figure 2, we see that for ImageNet and PACS, the calibration metrics behave like for CIFAR-10 and CINIC-10, matching the described behavior in the main text. We use 5 models from each of the enumerated architectures to build an ensemble of 25 models. Individual architectures also behave as expected as we ablate in Figure 5.

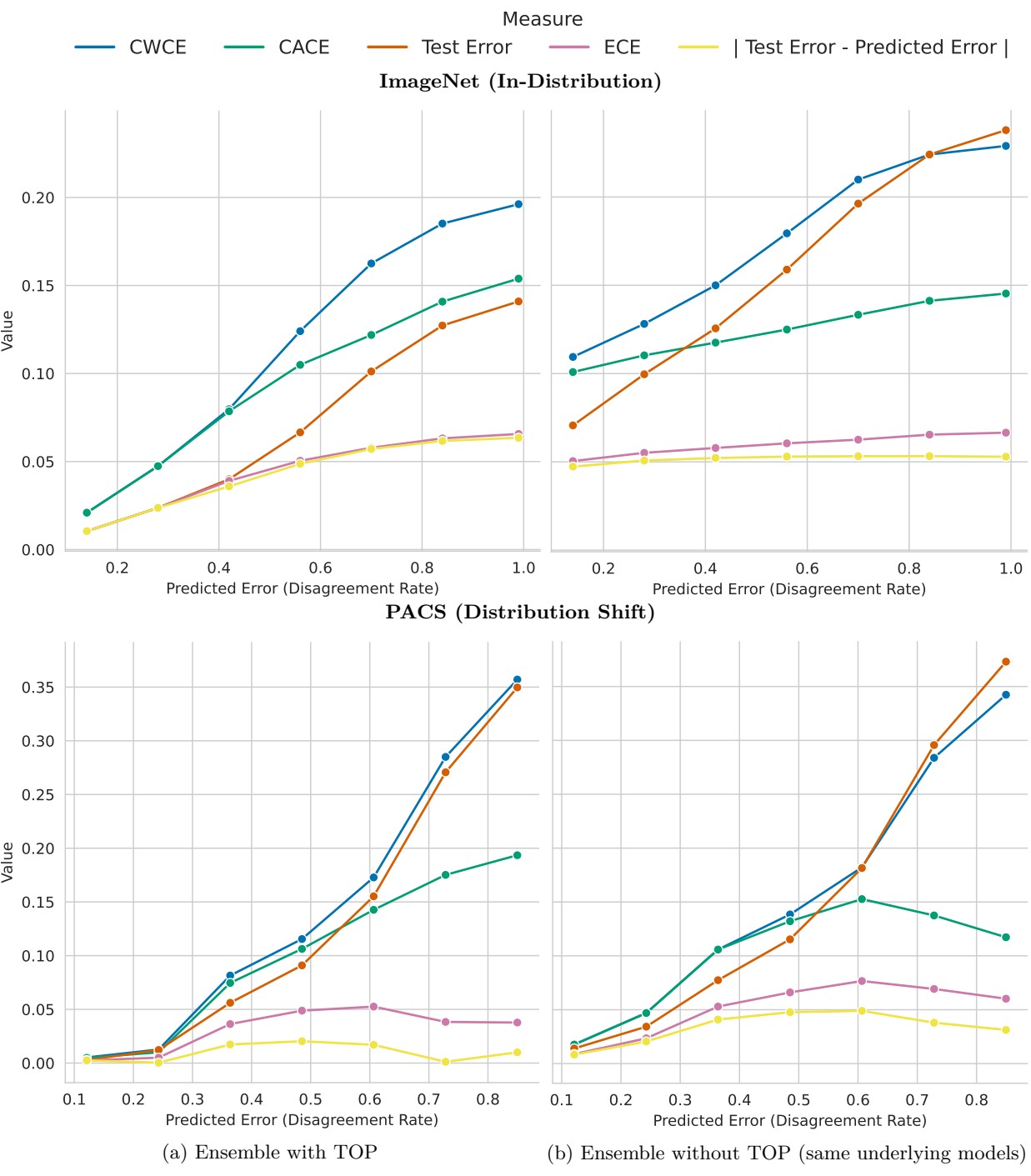

Figure 2: *Rejection Plot of Calibration Metrics for Increasing Disagreement In-Distribution (ImageNet) and Under Distribution Shift (PACS 'photo' domain → other domains).* Different calibration metrics ($ECE$, CWCE, CACE) vary across ImageNet and PACS' 'art painting', 'cartoon', and 'sketch' domains across an ensemble of 5 models trained on ImageNet and 25 models fine-tuned on PACS' 'photo' domain, depending on the rejection threshold of the predicted error (disagreement rate). Again, calibration cannot be assumed constant for in-distribution data or under distribution shift. The mean predicted error (disagreement rate) is shown on the x-axis. **(a)** shows results for an ensemble using TOP (following Jiang et al. (2022)), and **(b)** for a regular deep ensemble without TOP. Details in §E.2.

Additionally, in Figure 3 and Figure 4, we also show rejection plots using the Expected Information Gain/BALD for thresholding. We observe similar trajectories. Comparing these results with Figure 1 and Figure 2, we see that both the predicted error and the Bayesian metric behave similarly. We hypothesize that this could be because the datasets only contain few samples with high aleatoric uncertainty (e.g. noise), which would otherwise act as confounder (Mukhoti et al., 2021). See also the discussion in §6.

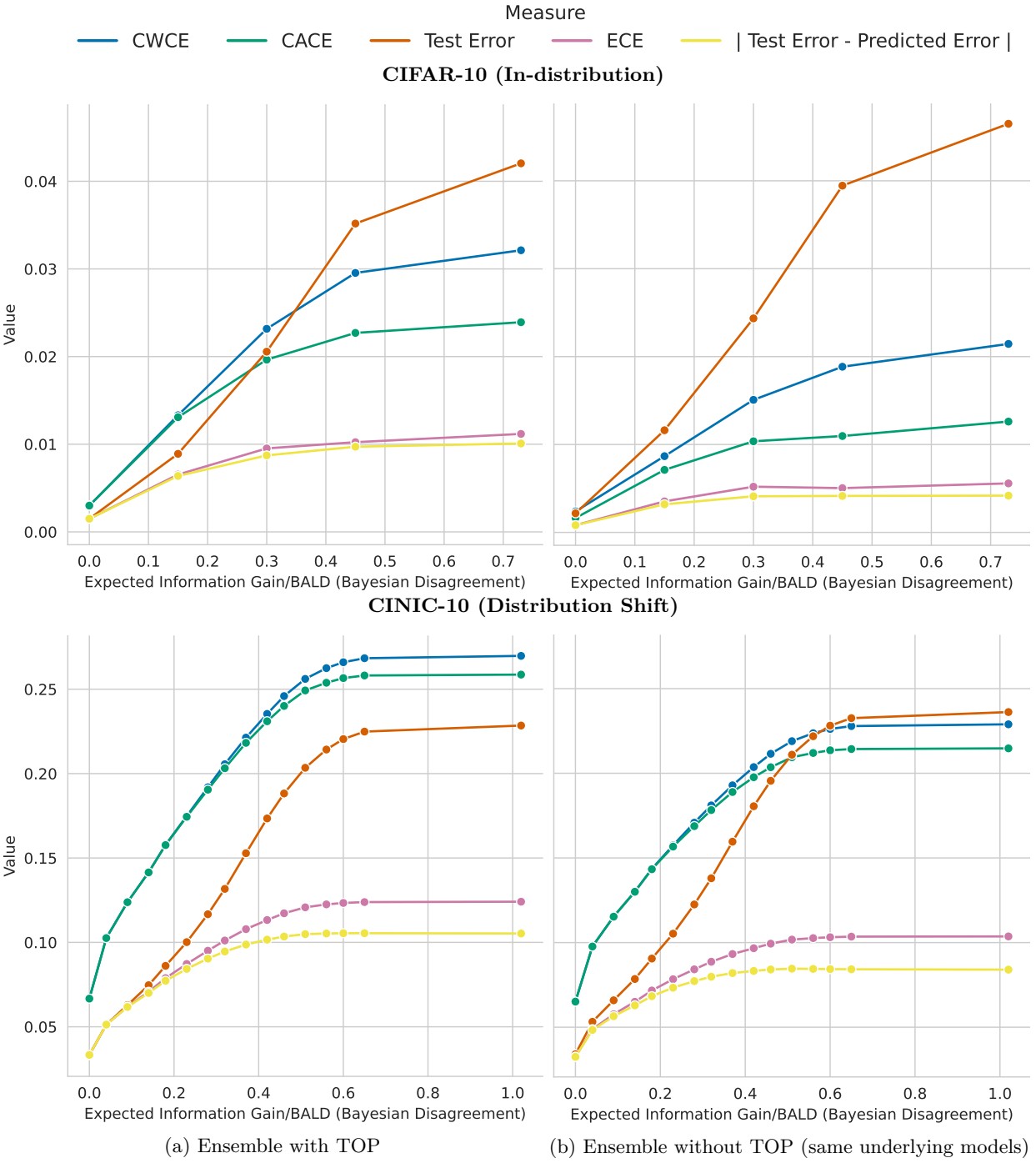

Figure 3: *Rejection Plot of Calibration Metrics for Increasing Bayesian Disagreement In-Distribution (CIFAR-10) and Under Distribution Shift (CINIC-10).* Different calibration metrics (*ECE*, CWCE, CACE) vary across CIFAR-10 and CINIC-10, depending on the rejection threshold of Bayesian disagreement (Expected Information Gain/BALD). The trajectory matches the one for prediction disagreement. We hypothesize this is because there are few noisy samples in the dataset which would act as a confounder for prediction disagreement otherwise. Details in §E.2.

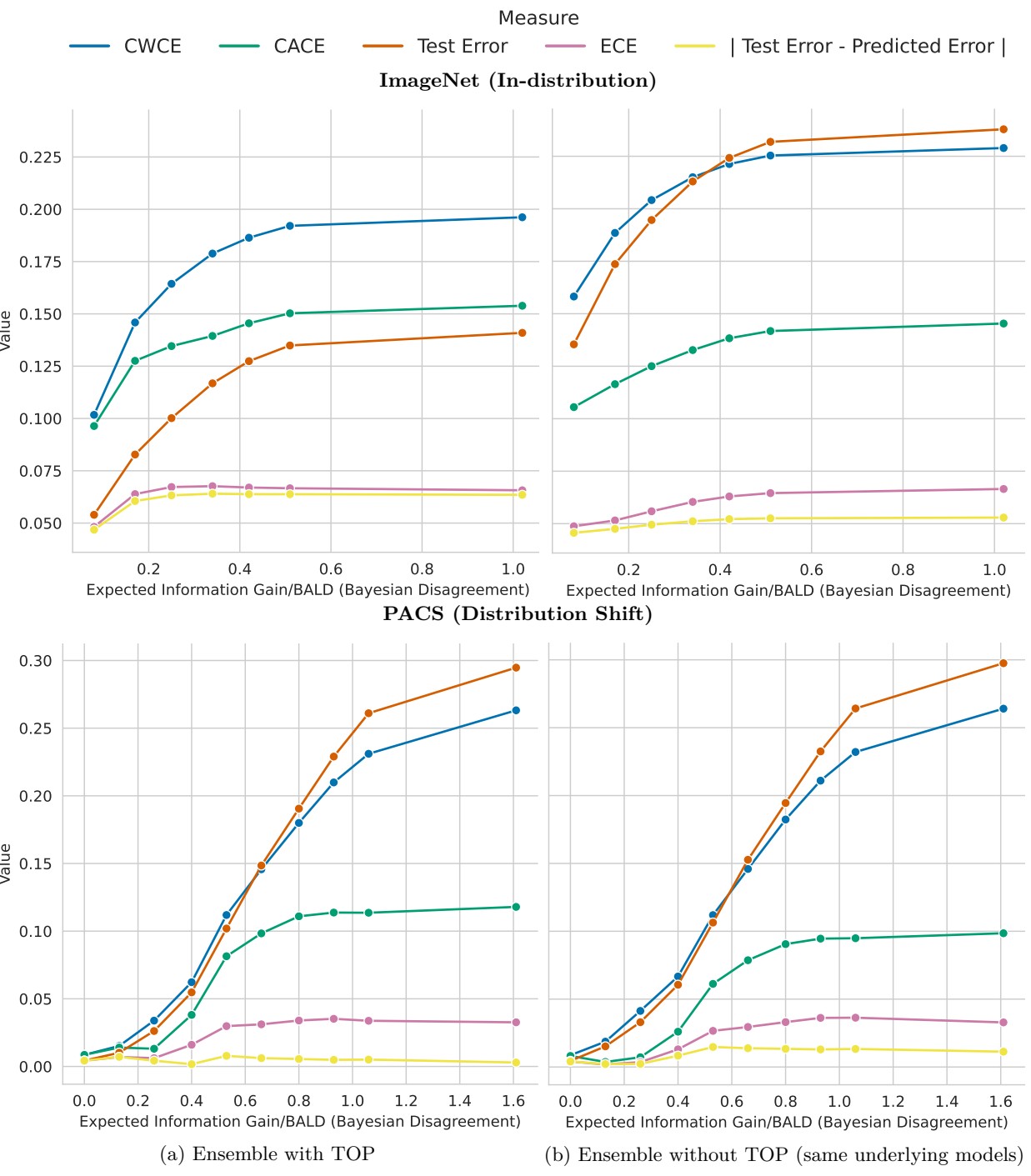

Figure 4: *Rejection Plot of Calibration Metrics for Increasing Bayesian Disagreement In-Distribution (CIFAR-10) and Under Distribution Shift (CINIC-10).* Different calibration metrics (*ECE*, CWCE, CACE) vary across CIFAR-10 and CINIC-10, depending on the rejection threshold of Bayesian disagreement (Expected Information Gain/BALD). The trajectory matches the one for prediction disagreement. We hypothesize this is because there are few noisy samples in the dataset which would act as a confounder for prediction disagreement otherwise. Details in §E.2.

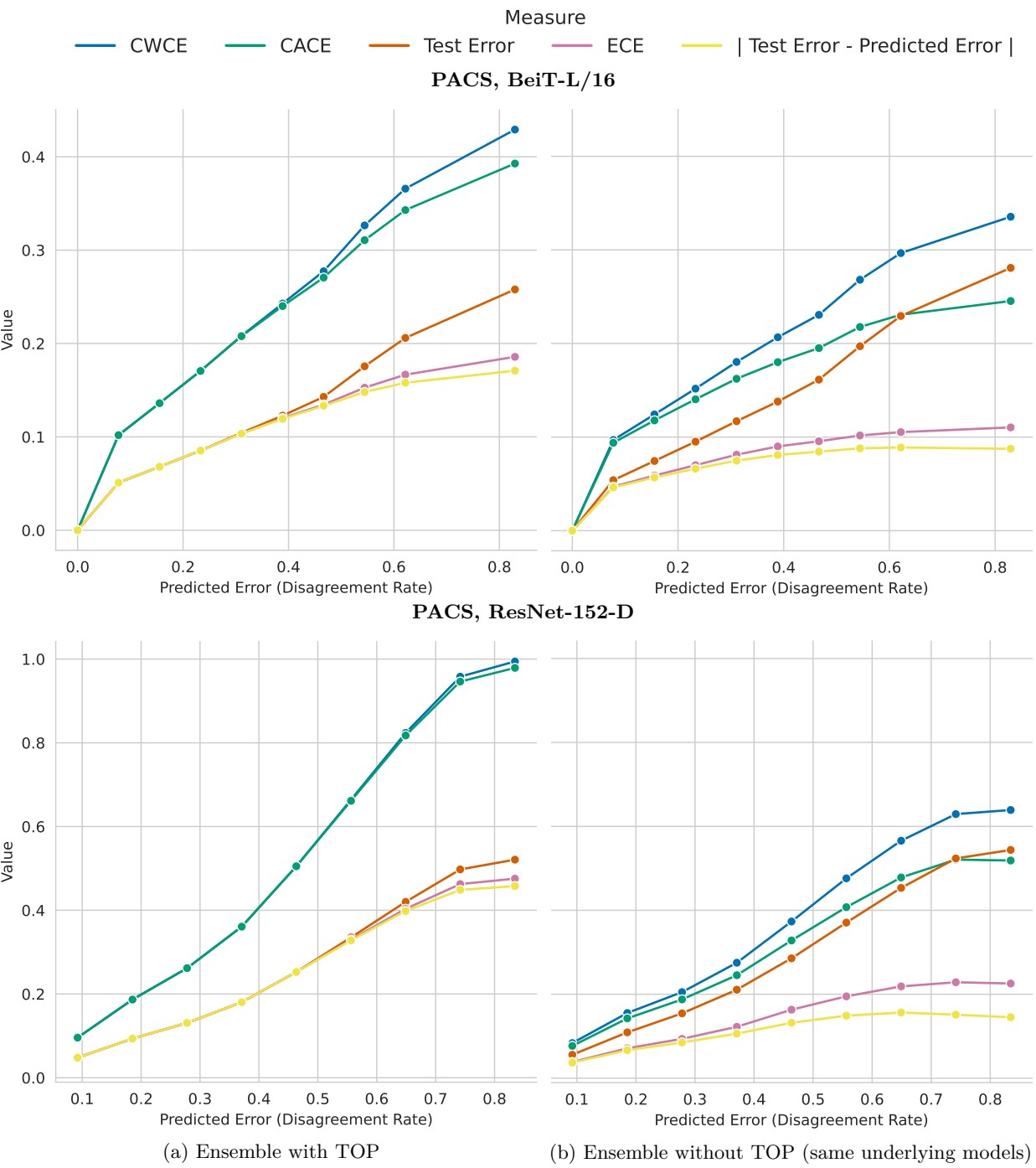

Figure 5: *Rejection Plot of Calibration Metrics for Increasing Disagreement Under Distribution Shift (PACS 'photo' domain → other domains) for Specific Model Architectures.* We use the same encoder weights and evaluate on an ensemble of 5 models, which were last-layer fine-tuned on PACS. We show ResNet-152-D and BeiT-L/16. Details in §E.2.

