# OpenReview forum: "A Note on "Assessing Generalization of SGD via Disagreement""
_TMLR — Accepted by TMLR_

### Review · Reviewer_AWck · 2022-08-29

**Summary Of Contributions:**

The work is a direct follow-up on Jiang et al. work, adding two important contributions. One is providing a Bayesian formulation of the original work, which allows to simplify the proofs. The other contribution is to argue that disagreement becomes unreliable as the value increases (or more exactly calibration becomes worse as disagreement increases) and calibration requires labels, making the use a disagreement instead of test error difficult.

**Broader Impact Concerns:**

I do not see any ethical concerns with this work.

**Requested Changes:**

Overall I recommend acceptance. I personally did not see any technical issues with the proofs. To a certain extent most of the theoretical work is just presenting the results of the original work from a Bayesian perspective, so the results themselves are not proposing something new, so in some sense we expect these theorems and lemmas to hold.
That said I cannot say I fully internalized the proofs, but I'm fairly confident they are ok.

Regarding the deterioration of the calibration, this result is mostly empirical. I think this observation is very useful, and I'm happy the authors have emphasized it in their work.
I think there is one change I would condition the acceptance on, which is for the authors to provide in the final version a bit more details on the experimental setup. I think the description provided is not sufficient for reproducing their results, and I can easily see an appendix where they can just enumerate all the details of their setup. While I think this requires little work, and the results agree with my intuitions so I have no doubt to distrust the empirical evidence, I think it is highly important to be fully transparent when it comes to empirical results.

Additionally it would be great if the authors can provide any kind of intuition of why this deterioration happens (from a more theoretical stand point). But of course I agree this might be a big ask, and formally it might be much harder to make this argument. I just felt at the end, even if informal, providing more intuition of what is happening under the hood would be great.

**Strengths And Weaknesses:**

I like the work a lot. I think it does a great job at presenting its argument clearly and offering a counter point to the original work.
Strengths:
 * very interesting results, offering significant context to the original work
 * well written, easy to follow
 * to the extent I followed the proofs (and I can't say I properly internalized them) I do not see any error with them

Weaknesses:
  * while the authors did a great job with the write-up, I think some parts of the proof can still be a bit hard to follow. I don't know if this really qualifies as a weakness, but personally I found that I can't just read the paper, but I have to go back and forth over the derivations to make sure I understand them correctly. I think this was partially because I was not familiar with the original work, and some of the quantities used in the work (e.g. predictive accuracy or error, which initially I thought I understood, but as going through the proofs I realized I had to go back to the definition and digest what they mean). That said, for a theoretical work, I do not know if this can be avoidable.

---

> ### Author Response · Authors · 2022-09-28
> **Response to Reviewer AWck**
>
> Dear Reviewer AWck,
>
> Thank you so much for your review! We are very happy that you like our paper and think it presents its argument clearly overall and presents significant context to the original work.
>
> We are also very grateful for your feedback, and we have used it to improve the paper in the following ways:
> * We have updated the paper to add an example for accuracy and predicted accuracy vs the top1 accuracy and predicted accuracy.
> * The experimental setup is described in much greater detail in appendix E1 now. We have also included relevant code & data in a supplementary archive, incl the scripts to recreate the plots. This should simplify reproductions. (For the WRN models, we used the training scripts from https://github.com/omegafragger/DDU.)
> * We have added another short section that hypothesizes as to the deterioration: the predicted error measures total uncertainty (= epistemic + aleatoric uncertainty). Because there are few noisy samples in the curated datasets we use, the predicted error is well aligned with the epistemic uncertainty - otherwise those noisy samples would be confounded with samples with high epistemic uncertainty and the calibration metrics would appear more stable as the model is reliable for noisy samples it understands. In that case, rejection plots along epistemic uncertainty would show a deterioration but along predicted error (as total uncertainty) not so much.
> * We have also added two sections to the appendix that look at the results from an information-theoretic perspective and additional experiments on ImageNet and PACS with different model architectures.
>
> Please let us know what you think and thank you so much for engaging with our work!
>
> Best wishes,\
>  The Authors

---

### Review · Reviewer_Tz6j · 2022-09-03

**Summary Of Contributions:**

This paper provides an alternative perspective to (Jiang et al. 2022), revisiting their theoretical and empirical results through a probabilistic lens. In particular, they simplify their results (which claim that looking at model disagreements can provide an upper bound on their test errors) by examining various definitions of (Bayesian) model calibration, propose new connections to prior works that emerge as a consequence, and demonstrate through experiments on CIFAR-10/CINIC-10 that the empirical findings in (Jiang et al. 2022) should be interpreted with care.

**Broader Impact Concerns:**

The authors did not include a broader impacts statement, though there do not seem to be any immediate ethical concerns or implications of this work.

**Requested Changes:**

Major:
- At least one more verification of a (dataset, architecture) combination to ensure that the empirical findings in the submission generalize to different settings.

Minor:
- In the introduction, there are several notions of uncertainty that would be helpful to be unpacked a bit further. For example, on page 2, what does “overall uncertainty” refer to in the context of epistemic/aleatoric uncertainty?
Nitpick: Notation section is a bit verbose.

Typos:
- “And thus includes a distribution [shift]” on page 10


**Strengths And Weaknesses:**

Strengths:
- The paper is clearly written and easy to follow. I particularly liked how this perspective allowed for more simplifications of (Jiang et al. 2022).

Weaknesses:
- Perhaps this isn’t the point of the submission (since it seems to be more of an explanatory paper), but do the empirical findings on CIFAR-10 hold for some of the other datasets or architectures explored in (Jiang et al. 2022)? Say, SVHN or PACS. I don’t think it’s necessary to reproduce all combinations of them, but it’d be more compelling to have more than one (dataset, architecture) example in the submission to drive the point home.

---

> ### Author Response · Authors · 2022-09-28
> **Response to  Reviewer Tz6j**
>
> Dear Reviewer Tz6j,
>
> Thank you so much for your review! We are very happy you like our paper and find that it is clearly written and easy to follow! Providing a different perspective and more simplifications was our goal, and we are happy you think so, too!
>
> We are also very grateful for your actionable feedback. We have used it to improve the paper, and we hope that the changes made satisfy your requests:
> * We have added additional verification with the PACS dataset (for distribution shift) and ImageNet (for in-distribution). We use pretrained models from the timm package for this. The results match the results in Figure 1.
> * We have clarified the notions of uncertainty, especially “overall uncertainty” as the sum of aleatoric and epistemic uncertainty, and used that to hypothesize why we see a performance deterioration as uncertainty increases.
> * (And we have fixed the typo which you have pointed out.)
>
> Thank you so much for your review and please let us know what you think.
>
> Best wishes,\
>  The Authors

---

### Review · Reviewer_RrY3 · 2022-09-14

**Summary Of Contributions:**

This short paper builds closely on and around the findings of Jiang et al. (2022), and makes several contributions: (1) The authors show that the results of Jiang et al. (2022) could be re-derived in a more straightforward manner through a Bayesian view. (2) The authors argue that a lot of the heavy lifting in the results are indeed due to the calibration assumption, which itself is not verifiable without labels leading to a circular argument; (3) The authors make a case that one should be cautious about interpreting the results when dealing with arbitrary distribution shifts, as the calibration properties may not generalize when distributions shift (and provide numerical evidence for it). Overall, I enjoyed reading the paper and liked the exposition of it.

**Requested Changes:**

Please reformat the paper for the technical derivations/exposition to become standalone, and then make connections with (Jiang et al 2022) once all technical results are derived/presented.

**Strengths And Weaknesses:**

The main strength of the paper is that it makes several contributions that shed light on understanding generalization via disagreement, as summarized above. The exposition of the paper is nice, the equivalent definitions to (Jiang et al 2022) are more intuitive, and the proofs are straightforward and understandable.The flow of the paper is generally clear, while I have a suggestion for improving it.

The main weakness of the paper is that the paper is written as if this is a response to (Jiang et al. 2022). My suggestion would be for the authors to make this a standalone paper, i.e., put down the definitions and assumptions, and do all the derivations and proofs; then in a separate section connect everything back to (Jiang et al. 2022). In particular, this would increase the archival value of the paper, and improve its readability.

---

> ### Author Response · Authors · 2022-09-28
> **Response to Reviewer RrY3**
>
> Dear Reviewer RrY3,
>
> Thank you so much for your review! We have very happy that you enjoyed reading our paper and its exposition and that you also agree with us that the results in the original work can be re-derived in a more straightforward manner and the equivalent definitions we provide are more intuitive.
>
> We have tried to make the paper as standalone as possible while matching the definitions of Jiang. We have tweaked the text in the new revision, but there was not enough time to perform a full rewrite. We apologize for that. We have also expanded the appendix with additional connections to information theory and experiments on additional datasets/model architectures.
>
> We are very grateful for your change request, and we hope you like the changes overall. Please let us know what you think.
>
> Best wishes,\
>  The Authors

---

### Review · Reviewer_dFis · 2022-09-14

**Summary Of Contributions:**

The paper presents a response to Jing et al. (2022). First, it rederives the results from Jing et al. using a Bayesian instead of a hypothesis space approach. This simplifies many of the proofs. Second, it questions the key conclusions of Jing et al. The authors demonstrate empirically that a model's calibration can deteriorate as prediction disagreement increases---especially under distribution shift. This means that to apply the bounds, calibration must be measured on the validation data, which requires labels.

Overall I like the paper and feel it makes a good contribution and should be accepted.

**Broader Impact Concerns:**

N/A.

**Requested Changes:**

*Major*
- I would recommend using \mathbb{P} for the operator p[E] that is equivalent to \mathbb{E}[1_E].
- For p(.) that is used to denote a p.m.f. or p.d.f., I think it should be clearer what distribution is being used by settling on a single notation, i.e. use a subscript to denote the distribution: Eq. (3) would be p_{\hat{Y}|X=x}(\hat{y}) = \mathbb{E}_\Omega[ p_{\hat{Y}|X=x,\Omega}(\hat{y}) ] and p(\hat{Y}=Y|X) would become p_{\hat{Y}|X)(Y). This would also resolve the issue with p(\hat{Y} |X) as p_{\hat{Y} |X}(\hat{Y})
- I'm happy to be outvoted on this if the other reviewers disagree.

*Minor*
-The first sentence of the introduction is incorrect. I understand it's just preamble, but there is significant research on the reliability of machine learning models and the field does not "trust them blindly."

**Strengths And Weaknesses:**

**Strengths**

- The paper is well written and nice to read. It states its conclusions clearly (see Sec. 6) and provides evidence for them.
- The Bayesian formulation of Jing et al.'s results is more concise and clearer than the hypothesis space view.
- The experiments clearly support the paper's main argument.
- The proofs are straightforward and self-contained in the main text.


**Weaknesses**
- I found some of the notation unnecessarily confusing. This is mainly due to the use of round and square parentheses for the p operator. I feel that either only p[.]  should be used, or the way p(.) is used should have a single consistent notation. The notation p(.) is used to denote a p.m.f. or p.d.f. The expression like Eqs. (2) and (3) use lowercase versions of the random variables to indicate which distribution is being referred to. Whereas Eqs. (4) and (5) use an equals sign to indicate the which r.v.s the density comes from. This can be unclear for expressions like p(\hat{Y} = Y | X), which evaluates the conditional p.m.f. of \hat{Y} conditional on X at the random variable Y. The issue is further confused by expressions like p(\hat{Y} |X) in Eq. (23). It seems like this is supposed to be the conditional p.m.f. of \hat{Y} conditional on X evaluated at the r.v. \hat{Y}.
- There could be experiments on more datasets and model types. In general, I believe the main empirical claims of the paper, but the experiments are somewhat limited.

---

> ### Author Response · Authors · 2022-09-28
> **Response to Reviewer dFis**
>
> Dear Reviewer dFis,
>
> Thank you so much for your review and feedback! We are glad that you think the paper is well-written and easy to read, and it provides a concise presentation that is supported by the experimental results.
>
> We have used your feedback to update the paper. We have added additional experiments on ImageNet (in distribution) and PACS (distribution shift). For this (and due to a limited compute budget), we have used pretrained models from the timm package. For PACS, we have fine-tuned the models on a subdomain of PACS and then evaluated them on the others (as distribution shift). The results for these additional datasets and model architectures are as expected.
>
> We have changed $p[...]$ to $\mathbb{P}[...]$. Thanks for the suggestion! We have also tried to use the subscript notation. We like it very much, and it makes it obvious which distribution we use and what variables are substituted --- however, we ran into issues for more complex conditional expressions: e.g. $p(\hat Y = Y \mid p(\hat Y = k \mid X) = q)$ became difficult to read as $p_{\hat Y \mid p_{\hat Y \mid X}(k) = q}(Y)$.
>
> We have also updated the first sentence. Thanks for pointing that out!
>
> Please let us know what you think. Apologies for the delay. The changes took longer than we expected.
>
> Thanks so much, \
> The Authors

---

### Author Response · Authors · 2022-10-29
**Thanks so much for the reviews and constructive feedback**

Dear Reviewers and Action Editor,

Thanks so much for your kind reviews and constructive feedback. The interactions on TMLR have been a great and positive experience.

We have uploaded the camera-ready now and made the repository public.

We've fixed a few typos for the camera-ready and added two paragraphs on other related literature to the Discussion section.

Thanks and best wishes,\
the authors

---

### Decision · Action_Editors · 2022-10-13

**Recommendation:** Accept as is

**Comment:**

The authors built on a previous work from Jiang et al. (2022), adopting a simplifying viewpoint which makes the previous results clearer while demonstrating the empirical limit (e.g., under distribution shift) of this previous work statement. The authors have followed the reviewers feedback and improved notations to make the paper more readable and added experiments to further support their claim.

**Audience:**

This submission is discussing generalization evaluation, distribution shifts, and calibration, which is of interest of a significant amount of machine learning researchers and practitioners.

**Claims And Evidence:**

The claims in this paper are supported by accurate, convincing and clear evidence: the theoretical claims have straightforward proofs that were verified by some competent reviewers and they have also been shown to occur empirically across different datasets and architectures.